# Thermal sensitivity of $CO_2$ and $CH_4$ emissions varies with streambed sediment properties

Sophie A. Comer-Warner [1], Paul Romeijn[1], Daren C. Gooddy [2], Sami Ullah[1], Nicholas Kettridge[1], Benjamin Marchant[2], David M. Hannah [1] & Stefan Krause[1]

Globally, rivers and streams are important sources of carbon dioxide and methane, with small rivers contributing disproportionately relative to their size. Previous research on greenhouse gas (GHG) emissions from surface water lacks mechanistic understanding of contributions from streambed sediments. We hypothesise that streambeds, as known biogeochemical hotspots, significantly contribute to the production of GHGs. With global climate change, there is a pressing need to understand how increasing streambed temperatures will affect current and future GHG production. Current global estimates assume linear relationships between temperature and GHG emissions from surface water. Here we show non-linearity and threshold responses of streambed GHG production to warming. We reveal that temperature sensitivity varies with substrate (of variable grain size), organic matter (OM) content and geological origin. Our results confirm that streambeds, with their non-linear response to projected warming, are integral to estimating freshwater ecosystem contributions to current and future global GHG emissions.

[1] School of Geography, Earth and Environmental Sciences, University of Birmingham, Edgbaston, Birmingham B15 2TT, UK. [2] British Geological Survey, Maclean Building, Wallingford, Oxfordshire OX10 8BB, UK. These authors contributed equally: Sophie A. Comer-Warner, Paul Romeijn. Correspondence and requests for materials should be addressed to S.A.C-W. (email: sxc469@bham.ac.uk)

Carbon fluxes from freshwaters, particularly rivers and streams, have been largely overlooked, as these were conceptualised as unreactive 'pipelines' transporting water from terrestrial to marine environments[1–4]. However, up to 50% of carbon is lost annually from inland waters through gas exchange as $CO_2$[5–7], returning 0.8 Pg of carbon directly to the atmosphere[1]. Previous research has mainly quantified surface water contributions[8–11] estimating global fluxes from streams and rivers to be 1.8 Pg $CO_2$-C $yr^{-1}$ [8], and 26.8 Tg $CH_4$-C $yr^{-1}$ [12]. Small streams appear to be particularly important, contributing ~15% of the $CO_2$ flux[8]. Although the $CH_4$ flux is dwarfed by that of $CO_2$, $CH_4$ fluxes may be regionally significant, and are seen to offset over 25% of the terrestrial C sink when considered as C equivalents[13–15]. Streambeds have been identified as 'hotspots' of carbon turnover[5,16–18], characterised by enhanced metabolic activity and nutrient spiralling[19–23]. GHG concentrations in streambeds are elevated with respect to surface waters, and concentrations between 71 nmol $CH_4$ $l^{-1}$[24] and 134 µmol $CH_4$ $l^{-1}$[25], and 130 µmol $CO_2$ $l^{-1}$[26] to 5 mmol $CO_2$ $l^{-1}$[27] have been observed in sediment pore-waters. Despite recent advances in analysing freshwater carbon cycling, the spatially and temporally variable drivers of enhanced GHG production in streambed sediments, $CH_4$:$CO_2$ ratios and the relative importance of sediment GHG to overall C emissions[2,5,7,28,29] remain insufficiently understood[1,5,7].

Temperature is the prime control of biogeochemical processing rates and a linear correlation between water temperature and GHG production has been observed between 0 and 65 °C[28,30]. Additionally, long-term warming has been shown to increase $CH_4$ emissions, and decrease $CO_2$ absorption and carbon sequestration in small ponds[31]. The impact of temperature on sediment C cycling and GHG production is likely to vary for streambed sediments of different geological origin, substrate, OM and nutrient content. Initial research suggests the potential of substantial GHG production from fine and organic carbon-rich sediments[28,32,33], but did not allow for any systematic analysis of sediment controls on GHG production. It is expected that temperature increases will be particularly important in agricultural lowland rivers and streams representing large areas of Europe, North America and Asia[34], which are characterised by excess nutrient loadings and OM-rich, fine sediments[12].

Herein, we investigate the temperature impacts on streambed sediment aerobic microbial metabolic activity (MMA) and GHG production, along a gradient of OM content in streambed sediments of different geological origin. Geological origin was investigated as different geologies are expected to have different GHG production rates. We incubated three substrates of differing grain size: Fine (silt-dominated underneath vegetation), medium (sand-dominated from unvegetated zones) and coarse (gravel-dominated from unvegetated zones) from the River Tern and River Lambourn - two agricultural, lowland, UK streams of contrasting geology (Sandstone and Chalk, respectively, Figure 1). Sandstone and carbonate were investigated as these are the dominant aquifer materials of the globe, therefore, allowing any conclusions to be more generally applied. Incubation experiments were performed in triplicate at 5, 9, 15, 21 and 26 °C, with the potential GHG production at 5 h investigated here. The smart tracer resazurin (Raz)-resorufin (Rru) system was used as a proxy for aerobic MMA, so that Rru production represented MMA (see methodology for a detailed explanation)[35]. GHG samples were collected from the headspace of the incubation jars, and so fluxes measured here include both diffusive and ebullitive pathways.

Here we demonstrate that microbial metabolic activity, and $CO_2$ and $CH_4$ production respond non-linearly exhibiting threshold responses to increases in temperature. Furthermore, the effect of temperature is highly variable depending on substrate, OM content and geological origin. This has large implications for GHG emissions from streams and rivers, with both temperature and OM content expected to change greatly under future climate and land-use scenarios.

## Results

**Microbial metabolic activity.** Details of the method and results of the statistical analysis are found in section 2.7 and Supplementary Table 1. MMA results are presented as ng of Rru produced per µg of Raz added to the jar at time zero. All reported errors represent ± 1 standard deviation. Where we refer to the testing of the contrasts detailed in Supplementary Table 1, we note the contrast number and $p$ value in parentheses.

MMA increased significantly with temperature across the non-control substrates (Fig. 2, C1, $p$ value < 0.01). MMA in Chalk$_{fine}$ increased by 1260% from 5 to 21 °C, producing MMA rates 400% larger than the second highest production (Chalk$_{medium}$ at 21 °C). The only other substantial activity was

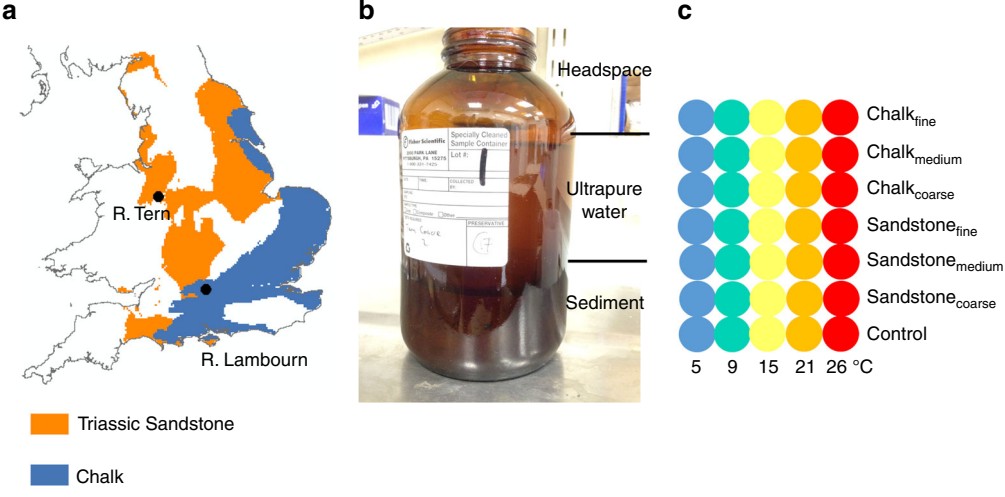

**Fig. 1** The location of the study rivers and experimental design. **a** Map of England and Wales, UK, showing the spatial distribution of Triassic Sandstone and Chalk aquifers, and the two study streams (River Tern and River Lambourn) [Contains British Geological Survey materials Copyright NERC [2016] and gadm.org]. **b** Incubation bottle used for the experiment, including a depiction of the distribution of sediment, water and headspace. **c** Experimental setup with each dot representing a triplicate of repeats

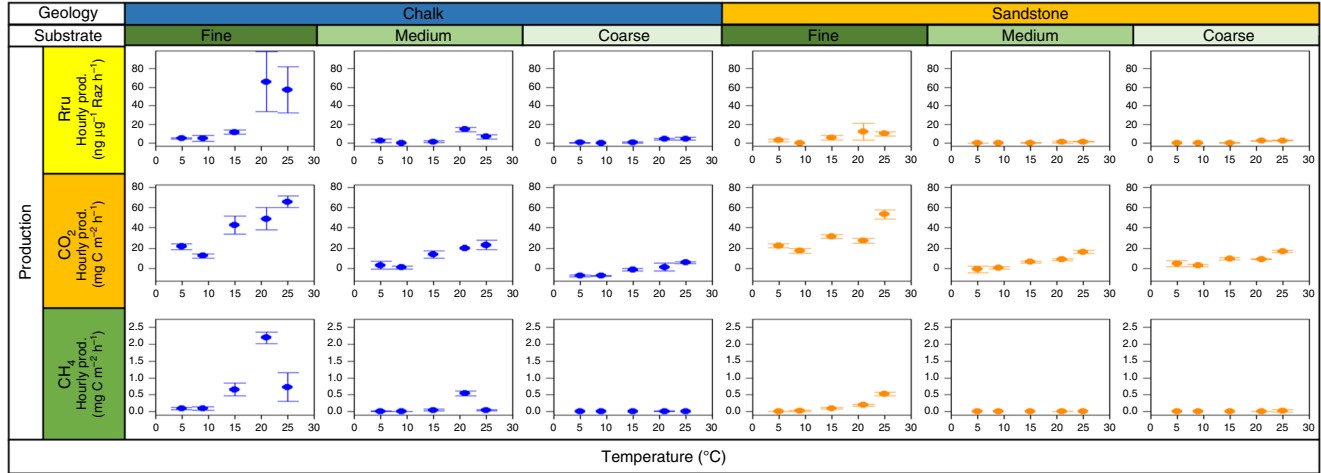

**Fig. 2** The effect of temperature on microbial metabolic activity and greenhouse gas production from stream sediments. Hourly production of Resorufin, $CO_2$ and $CH_4$ plotted against temperature for each substrate type (fine, medium and coarse) across the different geological origins (Chalk, blue, and Triassic Sandstone, orange). The error bars represent one standard deviation

observed in Sandstone$_{fine}$ and Chalk$_{medium}$ sediments at higher temperatures, resulting in significantly larger intersects for MMA in fine sediments compared to other sediment types for both geologies (C3, $p$ value < 0.01 for Chalk, and C4, $p$ value < 0.01 for Sandstone). There was also a significant increase in MMA with temperature in the fine sediments, compared with the control experiments (C5, $p$ value < 0.01). This was expected as fine sediments are predicted to have greater activities and there was a significant temperature-gradient across all sediment types. This was also the case for $CO_2$ and $CH_4$ production.

MMA was higher in Chalk$_{fine}$ than Sandstone$_{fine}$, despite the same OM content in both geologies (3.6%). We suggest that this is due to differences in the aromaticity of the carbon, which was 17.3% in Chalk$_{fine}$ and 20.9% in Sandstone$_{fine}$ sediments. Carbon in Chalk$_{fine}$ had a lower aromaticity, hence, the carbon was more bioavailable, producing greater MMA.

A linear relationship between temperature and MMA, as reported previously to be consistent across ecosystems[36], was not observed in this study. Greater microbial metabolism was observed in Chalk$_{fine}$, Chalk$_{medium}$ and Sandstone$_{fine}$ sediments at 21 °C than at 26 °C. Anaerobic conditions may cause a reduction in MMA; but water column oxygen concentrations for Chalk$_{fine}$, Chalk$_{medium}$ and Sandstone$_{fine}$ sediments increased between 21 °C and 26 °C, thus oxygen concentration cannot explain the observed decrease in metabolism (Supplementary Fig. 1).

OM content and geological origin had a substantial impact on MMA at higher temperatures (Fig. 3). Most sediments exhibited low rates of MMA between 5 and 15 °C, with only Chalk$_{fine}$ producing larger rates at 15 °C (11.8 ± 2.2 ng Rru µg$^{-1}$ Raz h$^{-1}$), indicating an onset of increased reactivity at 15 °C in Chalk$_{fine}$. The intersect for MMA was significantly larger for Chalk than Sandstone (C2 $p$ value = 0.02) and MMA was greater in the Chalk sediments at 21 and 26 °C. The difference in rates between geological origin was greater at 21 than 26 °C reflecting the higher MMA at 21 °C (maximum of 65.7 ± 32.1 ng Rru µg$^{-1}$ Raz h$^{-1}$) than 26 °C (maximum of 56.8 ± 24.7 ng Rru µg$^{-1}$ Raz h$^{-1}$) in the Chalk sediments. The metabolic rate was greatest in the fine sediments at higher temperatures, with the Chalk$_{fine}$ greater than the Sandstone$_{fine}$ at 21 and 26 °C.

Raz and Rru are known to sorb to sediments, which bears the risk that MMA could have been underestimated during these experiments, and all previous applications of the tracer. However, as such potential underestimation would have occurred throughout all investigated sediment types, it is expected that the impact on the interpretation of the results would be minimal. Additionally, it is expected that sorption would be slightly greater, and hence the underestimation greater, in the fine sediments. As the fine sediments had the greatest MMA, the potential underestimation in those sediments would not affect the conclusion that metabolism was greatest in the fine sediments. It is also possible that the sorption and mass recovery of Rru differed between the sediment types and geological origins, however, based on previous research[37] and the relatively low organic matter contents across all sediment types in this study, this is expected to have a negligible effect.

Differences in small production rates produced unrealistic temperature coefficient ($Q_{10}$) values, therefore, only $Q_{10}$ values where notable activity rates were observed, are discussed herein. This is also the case for the $CO_2$ and $CH_4$ discussions below.

$Q_{10MMA}$ values generally ranged between 0.0 and 3.3 (Supplementary Table 2), confirming previously reported values[34,38]. High $Q_{10MMA}$ values of 9.0 between 9 and 15 °C in Chalk$_{fine}$, 22.2 between 15 and 21 °C in Chalk$_{fine}$, and 1425.3 between 15 and 21 °C in Chalk$_{medium}$ sediments, reflected the elevated respiration rates observed in these sediments at the given temperatures. The observed $Q_{10MMA}$ value for Chalk$_{medium}$ between 15 and 21 °C was substantially larger than previously reported values (above), which is due to insignificant rates of MMA at 15 °C, followed by high rates at 21 °C once the microbial community responded to the elevated temperature. These $Q_{10MMA}$ values highlight the difference in temperature response between geological origin, with large increases in respiration rates only observed in the Chalk sediments at higher temperatures.

**Carbon dioxide production.** Details of the results of the statistical analysis are found in Supplementary Table 3. Chalk$_{fine}$ and Sandstone$_{fine}$ yielded the highest $CO_2$ production rates, with a 220 and 150% increase from 5 to 26 °C, respectively, with a maximum potential $CO_2$ production rate of 65.6 ± 5.9 mg C m$^{-2}$ h$^{-1}$ observed in the Chalk$_{fine}$ sediment at 26 °C. $CO_2$ production rates increased significantly with temperature across the substrates compared to the controls (Fig. 2, C1, $p$ value < 0.01). These results accord with the findings of previous research, which showed $CO_2$ production increased linearly with rising temperatures[28]. However, the relationship between temperature and potential $CO_2$

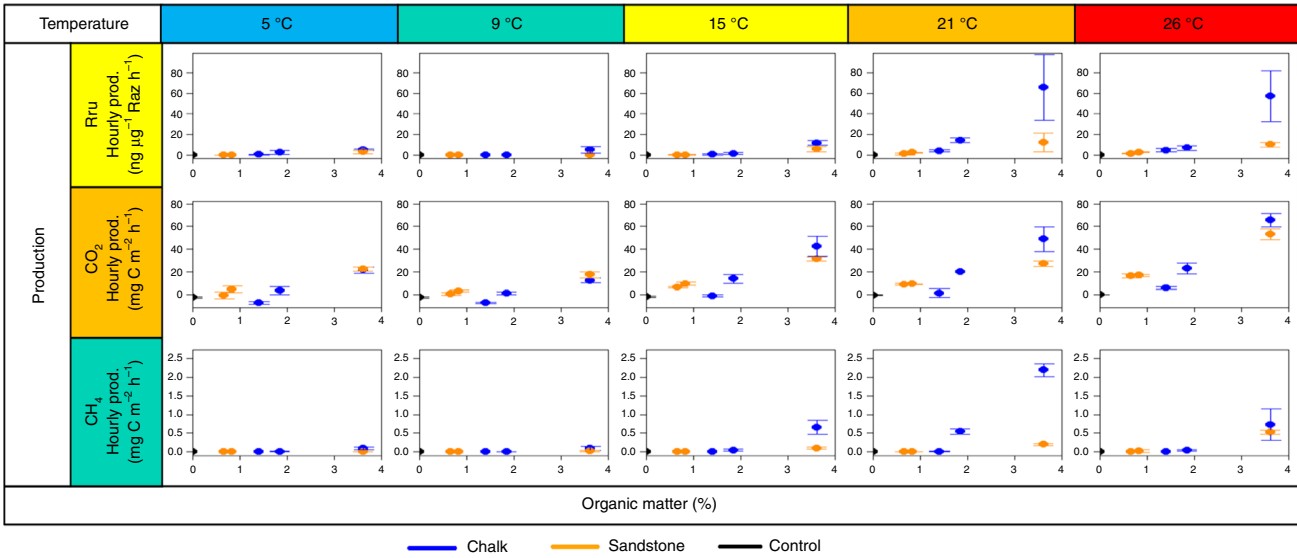

**Fig. 3** The effect of organic matter on microbial metabolic activity and greenhouse gas production from stream sediments. Hourly production of Resorufin, $CO_2$ and $CH_4$ plotted against organic matter content for each temperature (5, 9, 15, 21 and 26 °C) for Chalk (blue), Triassic Sandstone (orange) and control (black) experiments. The error bars represent one standard deviation

production observed in this study varied with substrate, with significantly (C3, *p* value = 0.01 for Chalk, and C4, *p* value < 0.01 for Sandstone) greater production rates in fine sediments than in medium and coarse sediments. The $CO_2$ production rates did not vary greatly between Chalk$_{fine}$ and Sandstone$_{fine}$ sediments, which was expected given the similar OM content between these sediments (3.6%). However, the MMA was greater in Chalk$_{fine}$ than Sandstone$_{fine}$; it has been noted that the $CO_2$ production rates did not mirror the Rru production rates observed, despite Rru production often used as a proxy for MMA. Somewhat surprisingly, in Chalk$_{coarse}$ at 5 and 9 °C, negative production of $CO_2$ was observed under aerobic conditions, which may be due to two processes. Firstly, negative production of $CO_2$ may occur at a high pH due to the aqueous carbonate system acting as a $CO_2$ sink. This has been observed previously at pH greater than 8.5[39], and the average pH of Chalk$_{coarse}$ at 5 and 9 °C was 8.3 ± 0.1, close to this threshold. It is possible, therefore, that the $CO_2$ produced here did not diffuse into the headspace, but remained in solution as carbonate. Secondly, the production rates were calculated from the difference in concentration between 0 and 5 h. As Chalk$_{coarse}$ had such low headspace $CO_2$ concentrations, it is possible that $CO_2$ from the atmosphere dissolved into the water column at these low temperatures, reducing the $CO_2$ concentration in the headspace over time.

OM content and geological origin had a substantial influence on potential $CO_2$ production rates, particularly at higher temperatures (Fig. 3). Both streams had similar rates of $CO_2$ production for fine sediments, which significantly exceeded production rates in medium and coarse sediments (C3, *p* value < 0.01 for chalk, and C4, *p* value < 0.01 for sandstone). Slightly higher $CO_2$ production rates were found in the medium and coarse sediments of Sandstone than in Chalk, with an OM content below 2%. In general, $CO_2$ production gradients were significantly greater in Chalk than Sandstone sediments (C2, *p* value < 0.01). Note that the intersect for Sandstone was significantly larger than that for Chalk (C2, *p* value = 0.02) indicating that the relationship with temperature was not linear and when the mean emissions per jar were considered, the difference was not significant. At 15 °C, Chalk$_{fine}$, Chalk$_{medium}$ and Sandstone$_{fine}$ responded to increased temperature with

enhanced production rates compared to 9 °C, indicating a threshold of 15 °C was required to enhance biogeochemical processing in both geological settings. At 21 °C, $CO_2$ production rates were similar to those at 15 °C; with no substantial increase in production observed. Large increases in $CO_2$ production rates were observed at 26 °C in the fine sediments to 65.6 ± 5.9 mg C m$^{-2}$ h$^{-1}$ for Chalk, and 53.1 ± 4.7 mg C m$^{-2}$ h$^{-1}$ for Sandstone. This is in contradiction to the observed MMA, which may indicate there is no metabolic reason for the observed decrease in MMA from 21 to 26 °C. The difference between $CO_2$ production rates in the fine sediments decreased between 21 and 26 °C.

$Q_{10CO2}$ values generally ranged between 0.1 and 4.0 (Supplementary Table 2), which were similar to previously reported values in lake sediments[40]. Higher values indicated a greater temperature dependency in some cases, with $Q_{10CO2}$ values of 8.1 for Chalk$_{fine}$ between 9 and 15 °C, and 4.9 for Sandstone$_{fine}$ between 21 and 26 °C. These $Q_{10CO2}$ values highlight the difference in response of the two geological origins, with Chalk$_{fine}$ reacting to temperature increases earlier than Sandstone$_{fine}$ (15 and 26 °C, respectively).

Potential $CO_2$ production rates ranging from approximately 70 to 156 nmol $CO_2$ g$^{-1}$ h$^{-1}$, from 10 to 25 °C in sandy sediments, and 147 to 261 nmol $CO_2$ g$^{-1}$ h$^{-1}$, from 3 to 22 °C in anoxic, fine sediments, have been observed previously[28,41]. These values for previous studies are greater than those observed here, where a range of 18 to 53 nmol $CO_2$ g$^{-1}$ h$^{-1}$ in Chalk$_{fine}$, and 18 to 45 nmol $CO_2$ g$^{-1}$ h$^{-1}$ in Sandstone$_{fine}$ from 5 to 26 °C were observed. Different units are discussed above due to a lack of available information to convert published values into mg m$^{-2}$ h$^{-1}$. The fine sediments were characterised by high carbon turnover rates of 82 days for Chalk and 105 days for Sandstone at 15 °C (representative of the current climate). When warming was considered these rates became 71 days for Chalk and 122 days for Sandstone at 21 °C, and 50 days for Chalk and 62 days for Sandstone at 26 °C. Our findings, thus, indicate that the turnover time for sediment C could almost halve under future climates where streambed temperatures reach 26 °C (representative to many Mediterranean streams), which bears severe consequences for biogeochemical cycling and nutrient spiralling in freshwater ecosystems.

**Methane production**. Details of the results of the statistical analysis are found in Supplementary Table 4. A maximum potential $CH_4$ production rate of $2.2 \pm 0.2$ mg C m$^{-2}$ h$^{-1}$ was observed in Chalk$_{fine}$ at 21 °C. $CH_4$ production rates did not increase substantially with temperature across all substrates (Fig. 2), which contrasts with observations of previous studies that report $CH_4$ fluxes to vary greatly with temperature across different ecosystems[30], although the rate of increase with temperature was significant across the sediment types compared to the controls (C1, $p$ value < 0.01). Instead, high $CH_4$ production was only observed in Chalk$_{fine}$, Chalk$_{medium}$ and Sandstone$_{fine}$ sediments, where production rates increased from 5 °C to 26 °C, this may be due to the suggestion that local factors are the dominant control on $CH_4$ production[12]. Ebullition may be responsible for this observation as it is related to finer sediments[42], explaining high production rates in Chalk$_{fine}$ and Sandstone$_{fine}$, and OM-rich sediments[42,43], resulting in elevated production rates in the three highest OM content sediments investigated here. The temperature sensitivity of $CH_4$ production rates was not consistent between the Chalk and Sandstone sediments. Rates were greater at 21 than 26 °C in the Chalk$_{fine}$ and Chalk$_{medium}$ sediments, whereas rates increased with temperature in Sandstone$_{fine}$ sediments. Patterns in Chalk$_{fine}$ and Chalk$_{medium}$ $CH_4$ production rates were consistent with trends observed in microbial metabolism and may be explained by an increase in $CH_4$ oxidation with temperature, alongside anoxic conditions[28]. A further explanation for the observed geological difference is that Chalk streams are thermally buffered due to groundwater, resulting in reduced thermal extremes in the summer and winter[44]; the Chalk microbial community is, therefore, expected to be adapted to lower temperatures, and may not respond well to extreme temperatures, e.g. 26 °C. Patterns in Sandstone$_{fine}$ $CH_4$ production rates, however, did not correspond to those of MMA, with a more obvious increase in $CH_4$ production than MMA with increasing temperature, which was non-linear.

Ratios of $CH_4$:$CO_2$ generally increased with temperature for fine sediments, with an increase in $CH_4$:$CO_2$ of 10% in Chalk$_{fine}$ and 2840% in Sandstone$_{fine}$ observed for a 1 °C temperature increase from 5 to 26 °C (Supplementary Fig. 2). The differences in increase in ratios between geological origin show that the relative increase was substantially higher in the Sandstone sediments than the Chalk sediments, indicating a larger proportion of C being released as $CH_4$ than $CO_2$ in the Sandstone sediments. The decrease in $CH_4$ production from 21 to 26 °C caused a decrease in $CH_4$:$CO_2$ in the Chalk$_{fine}$ sediment, resulting in an increase in ratio of 58% for a 1 °C temperature increase from 5 to 21 °C. An increase in $CH_4$:$CO_2$ ratio with temperature has been observed previously and is due to the high temperature dependence of $CH_4$ production, the variation between $CH_4$ and $CO_2$ kinetics, and the possible release of $CH_4$ from sediments before conversion to $CO_2$[28–30,45]. The ratios found herein are greater than those previously reported in fine, anoxic Chalk sediments, which found a 1 °C temperature increase produced a 4% increase in $CH_4$:$CO_2$ ratio, over a temperature range of 3–22 °C[28]. The medium and coarse sediments showed relatively little variation in $CH_4$:$CO_2$ ratio with temperature (between 4 and 15% with 1 °C increase from 5 to 26 °C), except at 21 °C in the Chalk$_{medium}$ sediment, which resulted in a 159% increase in $CH_4$:$CO_2$ ratio with 1 °C increase in temperature from 5 to 21 °C. Although most of the increases in medium and coarse $CH_4$:$CO_2$ ratios were relatively low compared to the fine sediments, they were still consistently higher (up to 4 times) than previously observed[28]. High $CH_4$:$CO_2$ ratios are frequently interpreted as indicators of significant human influence[12], which may explain the high values observed in this study for agricultural streams wherein $CH_4$:$CO_2$ ratios markedly exceeded values reported previously. The observed ratios highlight the relevance of investigating biogeochemical cycling to allow mitigation of GHG production, particularly in agricultural streams.

OM content and geological origin had a large influence on $CH_4$ production rates, especially at higher temperatures (Fig. 3), resulting in significantly different production rates between geologies (C2, $p$ value = 0.01). The fine sediments had similar production rates at both 9 °C and 26 °C. At 15 and 21 °C, an increase in production rates in the Chalk$_{fine}$ sediments resulted in a clear difference between the geological origins in the fine sediments. This indicates an onset of increased production rates at 15 °C in the Chalk$_{fine}$. There was a large decrease in $CH_4$ production rates in the Chalk$_{fine}$ sediments from 21 to 26 °C, alongside an increase in $CH_4$ production in Sandstone$_{fine}$, resulting in similar production rates at 26 °C (see above). Medium and coarse sediments had similar, low production rates across all temperatures, except at 21 °C where there was an increase in $CH_4$ production rate in the Chalk$_{medium}$ sediment to $0.548 \pm 0.075$ mg C m$^{-2}$ h$^{-1}$. This resulted in significantly greater production rate gradients and mean rates per jar in fine sediments in comparison to medium and coarse sediments (C3, $p$ value < 0.01 for Chalk, and C4, $p$ value < 0.01 for Sandstone). The significantly larger intersects for sandstone in these comparisons is indicative of a non-linear relationship.

48% of the $Q_{10CH4}$ values ranged between 0.0 and 4.1 (Supplementary Table 2), similar to previously reported values in lake sediments[40], with large values indicating a greater temperature dependency. Chalk$_{fine}$ sediments resulted in high $Q_{10CH4}$ values of 134.9 between 9 and 15 °C, and 9.3 between 15 and 21 °C. Sandstone$_{fine}$ resulted in high $Q_{10CH4}$ values of 227.6 between 9 and 15 °C, and 12.2 between 21 and 26 °C. These fine sediment, $Q_{10CH4}$ values, highlight the difference in response of the two geological origins, with both initially responding to increased temperature at 15 °C, then Chalk$_{fine}$ producing high $CH_4$ production rates earlier than Sandstone$_{fine}$ (21 and 26 °C, respectively). Chalk$_{medium}$ produced a high $Q_{10CH4}$ value of 163.2 between 15 and 21 °C, highlighting the elevated $CH_4$ production rate observed at 21 °C. The large $Q_{10CH4}$ values observed here were produced from initially insignificant rates of $CH_4$ production, followed by increased rates as the microbial community responded to increased temperatures.

$CH_4$ production rates have been observed to increase linearly with increasing temperature in anoxic, fine sediments of Chalk streams, increasing from 22 nmol $CH_4$ g$^{-1}$ h$^{-1}$ at 3 °C to 80 nmol $CH_4$ g$^{-1}$ h$^{-1}$ at 22 °C[28]. These rates were far greater than those observed here, which ranged from 0.1 nmol $CH_4$ g$^{-1}$ h$^{-1}$ at 5 °C to 0.6 nmol $CH_4$ g$^{-1}$ h$^{-1}$ at 26 °C in Chalk$_{fine}$, and 0.0 nmol $CH_4$ g$^{-1}$ h$^{-1}$ at 5 °C to 0.4 nmol $CH_4$ g$^{-1}$ h$^{-1}$ at 26 °C in Sandstone$_{fine}$. The low, similar $CH_4$ production rates observed here in the medium and coarse substrates across all temperatures, except in Chalk$_{medium}$ at 21 °C, (maximum of 0.03 nmol $CH_4$ g$^{-1}$ h$^{-1}$) has been observed previously, although previous experiments used seasonally collected sediments and so other environmental factors, such as substrate availability, may have also been affecting production rates. These previously observed rates were approximately 1.0 nmol $CH_4$ g$^{-1}$ h$^{-1}$ [41], higher than those observed here in the medium and coarse sediments, which may be due to the larger quantity of total organic carbon present in the sediments of the previous study.

**Dissolved oxygen**. In general, the oxygen concentration of the water column decreased with increasing temperature, resulting in a large difference between the lower and higher temperatures. The oxygen concentration of the water column showed a similar trend for both the Chalk and the Sandstone sediments, with similar

concentrations observed from 5 to 15 °C, before decreasing to similar concentrations at 21 and 26 °C, resulting in anaerobic conditions in some sediments. There was also a decrease in dissolved oxygen with increasing temperature in the controls, as was expected due to the physical controls over oxygen solubility, however, this decrease was not as pronounced in the controls as in the sediment treatments (Supplementary Fig. 1).

## Discussion

Increased temperature generally led to a rise in sediment respiration and GHG production in the investigated streambed sediments. Importantly, in contrast to previous studies, this relationship with temperature was non-linear for MMA and $CH_4$ production, with threshold responses observed[28,30,36]. Although Rru production has been developed as a proxy for aerobic microbial respiration, it is possible that Rru is produced in the jar alongside methanogenesis, due to oxic and anoxic sediments existing simultaneously[46]. This explains, therefore, the observations in some sediment types of simultaneous Rru, $CO_2$ and $CH_4$ production, especially at higher temperatures. Control experiments showed negligible production of metabolic activity and GHG production (Supplementary Fig. 3).

We found the temperature sensitivity of streambed sediment respiration and GHG production to be dependent on substrate, OM content and geological origin, with the greatest reactivity, and largest responses to increased temperature, found in the fine, high OM content sediments. This observation is likely explained by the large surface area provided by fine sediment, which leads to greater microbial populations[47]. Reactivity was generally greater in the Chalk than the Sandstone sediments; which resulted in Chalk_fine sediments being characterised by the highest rates of MMA and GHG production. GHG production in Chalk sediments responded to increased temperatures earlier than Sandstone sediments. Increased rates of sediment respiration and GHG production associated with fine, OM-rich sediments, as well as increased temperature, are consistent with previous research[28,30,32,33,36,41,42,48–50], with $CH_4$ results explained by the promotion of methanogenesis in fine, OM-rich sediments[12,28].

Laboratory incubation experiments, while allowing individual drivers to be investigated, may not compare one-to-one to in-situ conditions within the natural system where boundary conditions are less constrained (see section 2.8). As such, the results of this study provide potential rates of MMA and GHG production, which indicate the difference in magnitude of rates between sediments of varying substrate, OM content and geological origin, and differences in response to temperature due to these varying characteristics.

Our results demonstrate that biogeochemical processes in streambed sediments, such as respiration, are affected by temperature, substrate, OM content, and geological origin. The upscaling of GHG production, both temporally and spatially, should, therefore, consider the spatial variability of these confounding factors. When considering variations and trends in GHG production, increased temperature is a key driver of greater GHG production, and is expected to increase under future climates, global change and increased groundwater abstraction[12,29,51,52]. Fine, OM-rich sediments were also found to enhance GHG production, which are expected to increase due to land-use change and greater weathering rates[12]. Additionally, fine, OM-rich sediments introduce spatial variability in biogeochemical cycling, along with varying geological backgrounds. Of particular importance, therefore, are agricultural, lowland rivers, typically characterised by high nutrient loading and fine, OM-rich sediments[12,53]. The investigated sediments of agricultural rivers are representative for a wide range of ecosystems across much of Europe, North America and Asia, contributing significantly to the atmospheric burden of GHG.

Comparing the average potential $CO_2$ production rates from all sediments at 15 °C (representing typical present-day temperatures) to previous estimates, demonstrates an increase in stream and river $CO_2$ emissions to the atmosphere of 269% in this study[8]. If the streambed temperature was to increase to 21 or 26 °C this would result in a 329 and 552% increase in $CO_2$ flux, respectively, relative to current emissions estimates[8]. This rises to a 340 and 557% increase when methane is included, showing that substantial increases in C emission from streams are expected under future warming of streambed sediments up to 26 °C. These fluxes were calculated per COSCAT region 402 (Coastal Segmentation and related catchments of the UK, Ireland and Iceland). Comparisons between incubation experiment results and global observations allow the quantification of the effect of streambed GHG production, in relation to increasing temperature, on GHG efflux from streams and rivers. Incubation experiments and global observations represent different scales and natural conditions; therefore, these increases in effluxes represent an estimate of the influence of streambed GHG production at different temperatures and some caution needs to be exercised when predicting field fluxes under future climate change.

While the issues with upscaling incubation experiments to surface water emissions are appreciated, when used as an approximation (as above), the temperature-induced increase in $CO_2$ emissions identified in this study contradicts previous research that warming does not increase $CO_2$ emissions from surface waters[54]. Previous research has focussed on surface water only (not accounting for streambed sediment contributions) and noted that $CO_2$ emissions may be affected by temperature if ecosystem respiration and gross primary production are independent[54]. Our results indicate the importance of considering streambed respiration and subsequent $CO_2$ production, in C fluxes from streams and rivers, as this may alter the temperature-dependence of $CO_2$ emissions.

Future research on the environmental factors driving high GHG production rates in streambed sediments is required to fully understand stream carbon dynamics and emissions, and their response to future climates, to enable upscaling of GHG emissions to national and global scales. Research presented herein demonstrated that upscaling estimates of stream global carbon cycling in response to temperature must account for local and regional scale complexity in streambed geology and sedimentology. Our results highlight non-linearity and threshold responses of streambed GHG production in response to streambed warming. Furthermore, temperature sensitivity varied with substrate, OM content, and geological origin. Our results, therefore, demonstrate the importance of considering streambed production when estimating the contribution of freshwater ecosystems to global GHG emissions; especially due to observed non-linearity in streambed GHG production with increased temperature.

## Methods

**Sediment collection**. Sediment was collected from the top 10 cm of the streambed in the River Tern and the River Lambourn, UK in September 2015. To achieve a gradient of organic matter contents in the samples the sediment was collected from areas with the following characteristics: silt-dominated sediment underneath vegetation (type 1), sand-dominated sediment from unvegetated zones (type 2) and gravel-dominated sediment from unvegetated zones (type 3). The sediment was sieved (type 1 at 0.8 cm, and types 2 and 3 at 1.6 cm), to avoid large stones dominating a large proportion of the sediment within the incubation jar, and homogenised. The sediment was then stored air tight, in the dark, at 4 ± 1 °C, for 3 weeks before the incubation experiments began. Each incubation temperature treatment was performed in a separate week, over a period of 7 weeks. To minimise any potential effects of sediment storage the order of the temperature treatments

was randomised, rather than performed in sequence, e.g., from the lowest to highest temperature.

**Incubation experiments**. The incubations were performed with 300 ml sediment and 500 ml ultrapure water (18.2 MΩ) in pre-weighed 1 L amber glass jars (acid (10% HCl) and ultrapure water-rinsed) with lids with septa. Ultrapure water was used to allow sediment processes to be investigated independently of stream water solutes and microbes. Controls of 500 ml ultrapure water (18.2 MΩ) with no sediment were also prepared. The 3 substrate types from the 2 geological origins, as well as the control experiments, gave a total of 7 substrates, which were ran in triplicate resulting in 21 incubation jars per temperature treatment (Fig. 1). Once the samples were added the jars were placed, with lids ajar, into an incubation oven at treatment temperature for 3 days prior to the beginning of the incubation time period. The incubations were performed for 5 h at 5, 9, 15, 21 and 26 °C. Sampling occurred at 0 and 5 h as described below.

To enable the results from these incubation experiments to reflect in situ processes, real world conditions were emulated as closely as possible by manually swirling sediment slurries after the addition of the ultrapure water. This allowed for re-sorting of sediments with heavier particles at the base and finer particles settled on the top, to mimic natural sediment sorting and deposition conditions. Additionally, once incubated, the disturbance within the system was minimised to avoid any impact either on dissolution of headspace $O_2$ into the water and sediments, and the ebullition-based fluxes of GHG, particularly $CH_4$. This allowed for comparing fluxes from sediments of different texture and organic matter content, across the different geologies, under a constant sediment surface area. Thus, differences in sediment porosity constraints on oxygen diffusion were minimized to avoid experimental anomalies, and subsequent effects on production rates and $CH_4$:$CO_2$ ratios.

Prior to the start of the experiment a 15 ml water sample was taken and ran as a background sample on the fluorometer, this water was then returned to the correct incubation jar. One sample each for the Lambourn and the Tern were kept for later use in fluorometer calibration. At 0 h a jar was removed from the incubation oven, opened and 5 ml of (between 14.5 and 15.4 ppb) Raz solution was added to the water column and stirred. A 15 ml water sample was taken and filtered (0.45 μm ultrapure water-washed (18.2 MΩ) nylon) into the fluorometer to measure the Rru absorbance, after measurement the sample was returned to the incubation jar to maintain water volume. The fluorometer was rinsed with ultrapure water (18.2 MΩ) between samples. The oxygen concentration of the water column was measured (FireSting Fibreoptic DO probe), and the headspace of the jar was equilibrated with the surrounding air and the jar closed. The oxygen concentration of the headspace was measured with a needle probe (FireSting Fibreoptic DO probe) through the septa of the lid, and 2 × 15 ml gas samples were taken in a syringe (helium-rinsed) into pre-evacuated 12 ml exetainers. The jar was then placed back into the incubation oven. This procedure was repeated for all jars. After 5 h a jar was removed from the incubation oven, and gas samples taken as described above. The jar was then opened and the sampling procedure for t = 0 was followed. This procedure was repeated for all jars.

**Determination of sediment properties**. Following the incubation experiments the jars were weighed to give a wet weight and dried in an oven at 65 °C for 3 days, followed by 105 °C for 1 day. The jars were then weighed again to provide a dry sediment weight and a water weight for each jar. The dry sediment of 5 °C was used to determine the OM contents of the sediment types[55]. The sediment was sieved (2 mm), homogenised and a sub-sample of each jar sediment was placed into a pre-weighed crucible. The sample was dried at 105 °C overnight and weighed, resulting in subsamples of 14.8–24.8 g of dry sediment. The crucibles were then placed into a furnace at 550 °C for 6 h and then weighed to determine the OM content of the sediment.

**Aerobic microbial metabolic activity**. The Raz-Rru system has been developed as a reactive tracer due to the utilisation of Raz as an electron acceptor in aerobic respiration, resulting in the irreversible conversion of Raz to Rru[56,57]. Rru production can, therefore, be used as a proxy for aerobic microbial metabolic activity[56,57], Rru production is usually normalised by the amount of Raz detected in samples, to account for any losses of the tracer from the system. As we performed experiments in closed jars, we only measured Rru in the samples. The concentration of Rru in the samples were measured on a fluorometer (GGUN FL30 (Albilia Sarl, Switzerland)) as a proxy for aerobic microbial metabolism[56], with lamps set to detect the fluorescence of Rru[58]. Two fluorometers were used to ensure reading accuracy and the fluorometers were calibrated once a week with background water, Rru (100 ppb) and a mixture (50 ppb Raz, 50 ppb Rru). Fluorometer performance data was calculated using a 93.1 ppb standard of Rru, which resulted in an accuracy and precision of 0.4 and ± 0.7 ppb. The limit of detection for the GGUN FL30 fluorometers is 1 ppb for Rru[58]. The concentration of Rru in the measured samples ranged from 0.0 to 139.6 ppb, therefore, some samples were below the limit of detection (LOD) of the fluorometers. The maximum hourly production rate (dependent on the amount of Raz added to the jar) yielded from a Rru concentration of 1 ppb was 1.7 ng Rru μg$^{-1}$ Raz h$^{-1}$, and any samples below the LOD are presented here as actual values, accounting for some of the low MMA rates observed.

The design of the GGUN FL30 fluorometer allows a calibration with only 100 ppb Rru to be sufficient for these concentrations, here a calibration was performed with two Rru concentrations to improve data quality.

**Carbon dioxide and methane concentrations**. The concentration of carbon dioxide and methane in the gas samples was determined using an Agilent 7890A Gas Chromatograph (GC) - Flame Ionisation Detector (FID). The FID measures methane and so the carbon dioxide was methanised with a catalyst before passing to the FID. The GC had a 1 ml sample loop in a splitless orientation, and an oven temperature of 60 °C. The FID was set to 250 °C with a hydrogen flow of 48 ml min$^{-1}$, air flow of 500 ml min$^{-1}$ and a make-up nitrogen (pure) flow of 2 ml min$^{-1}$. The run time was 7 m and the gases eluted at 3.5 and 5.7 m for methane and carbon dioxide, respectively. Machine performance data were calculated using an external standard with concentrations of 1051 ppm $CO_2$ and 9.8 ppm $CH_4$. This resulted in an accuracy, precision and LOD of 13.4, ± 14.8 and 8.2 ppm, and 0.07, ± 0.11 and 0.15 ppm, for $CO_2$ and $CH_4$, respectively.

**Temperature coefficient values**. The temperature coefficient value ($Q_{10}$) quantifies the temperature dependence of a biological process and is here used to investigate the biological processes of MMA, and $CO_2$ and $CH_4$ production. The $Q_{10}$ value of a process is calculated using Eq. (1)[59].

$$Q_{10} = \left(\frac{\text{Process}_{T_2}}{\text{Process}_{T_1}}\right)^{\left(\frac{10}{(T_2 - T_1)}\right)}, \qquad (1)$$

where Process is the biological process under consideration at $T_1$ and $T_2$, and $T_1$ and $T_2$ are the respective temperatures at which Process was measured. Although $Q_{10}$ values are typically calculated where $T_1$ and $T_2$ are 10 °C apart, using the form of the $Q_{10}$ equation given above allows $T_1$ and $T_2$ to have different temperature intervals. $Q_{10}$ values were calculated between the different incubation temperatures, so that $T_1$ and $T_2$ were 5 and 9, 9 and 15, 15 and 21, and 21 and 26 °C (see Supplementary Table 2).

**Statistical inference**. We base our inference regarding the relationships between temperature, sediment type and MMA or GHG emissions upon the linear model:

$$y(s, t) = \sum_{i=0}^{6} \left\{ I_i(s)\alpha_i + I_i(s)\beta_i t \right\} + \varepsilon, \qquad (2)$$

where $y(s,t)$ is the measured MMA or GHG emission from sediment class, $s$, at temperature $t$, $I_i(s)$ is an indicator function that is equal to one when $i=s$ and is zero otherwise, the $\alpha_i$ and $\beta_i$ are model coefficients for the intersect and gradient terms and each $\varepsilon$ is independent and realised from a Gaussian distribution with zero mean and constant variance. Thus, each response curve is represented by a linear function. The seven sediment classes are: 0 – control, 1 – Chalk$_{fine}$, 2 – Chalk$_{medium}$, 3 – Chalk$_{coarse}$, 4 – Sandstone$_{fine}$, 5 – Sandstone$_{medium}$ and 6 – Sandstone$_{coarse}$.

We estimate the model coefficients $\mu = [\alpha_0,...,\alpha_6, \beta_1,...,\beta_6]^T$ for each GHG variable (where $T$ denotes the transpose) by residual maximum likelihood. Where the residuals that result are inconsistent with the Gaussian assumption, a shift and a log transform are applied to the data and the model is re-estimated until approximately Gaussian residuals are achieved.

We initially apply ANOVA tests to the null hypotheses that $\alpha_0 = \alpha_1 = ... = \alpha_6$ and $\beta_0 = \beta_1 = ... = \beta_6$. If these hypotheses can be rejected at the 0.05 level we then explore the factors causing variation in the $\alpha_i$ and/or $\beta_i$. We control the Type-1 errors across the experiment by testing a series of planned orthogonal contrasts, which are outlined in Supplementary Table 5. If appropriate, the same comparisons were applied to the $\beta_i$ gradient coefficients. The probability of achieving the estimated value of each contrast if the true value is zero, was then calculated by a Wald test following the methodology described by Lark and Cullis (2004)[60]. These pre-specified hypotheses and contrasts are based on the results from previous studies[28,30,36] that MMA and GHG emissions are linearly related to temperature. To confirm our findings in circumstances where this assumption is inappropriate, we also estimate a linear model of the mean $y(s,t)$ for each jar (i.e., averaged across all of the treatment temperatures) and again consider contrasts C1 to C5.

If either of the initial ANOVA hypotheses cannot be rejected then the corresponding $\alpha_i$ or $\beta_i$ in the model are replaced by a single constant coefficient and the remaining parameters are re-estimated before the above contrasts are tested.

**Experimental limitations**. The results of incubation experiments such as the one conducted here allow for systematic analyses of potential rates of MMA and GHG production. Since such systematic analyses are always based on model systems, one to one comparisons with site-specific in-situ measurements under less controlled conditions can remain challenging. Instead, systematic incubation studies support generalisations of the dependence of MMA and GHG production to varying temperature, substrate type, OM content and geological origin. Despite the fact that, like all methods, laboratory incubation experiments have limitations, they provide powerful tools to systematically investigate the impact of individual drivers

and remain the best way to control experimental treatment conditions that, in contrast to site-specific field studies, warrant generalisations of results.

In this study, the duration of experimental treatments required that sediment be stored over the duration of the entire experiment and mixed with ultrapure water instead of river water to guarantee stationarity of chemical conditions in the applied water source. The risk of this affecting the experimental results, e.g. by causing some alteration of the microbial community over the period of sediment storage, is considered minimal; however, the sediment was stored in cold and dark conditions and the order of temperature treatments were randomised to avoid any time-lapsed anomalies in the results as a function of temperature. The risk of critical dilution of pore-waters with ultrapure water that would have the potential to affect microbial communities is considered minimal since sediment and pore-water solute concentrations, such as nutrients, in the investigated lowland streams are high, with the ultrapure water addition likely to cause only minimal effects to the microbial communities that tend to be associated with sediment particles and so likely only experienced some dilution of pore-waters. This, combined with the large responses to temperature increases in the fine sediments that were observed in this study, indicate that the microbial communities were not greatly affected by osmotic shock or similar processes.

**Data availability**. The data that support the findings of this study have been deposited in the NERC Environmental Information Data Centre, entitled 'Carbon dioxide and methane emissions from incubated streambed sediments from the rivers Tern and Lambourn, England (2015)' (https://doi.org/10.5285/3a0a5132-797c-4ed5-98b9-1c17eaa2f2b7; https://catalogue.ceh.ac.uk/documents/3a0a5132-797c-4ed5-98b9-1c17eaa2f2b7).

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

## Acknowledgements

The authors would like to thank the Natural Environment Research Council (NERC) for their funding of this project through a Central England NERC Training Alliance studentship (grant number 1602135), as well as the European Union for their funding through the Seventh Framework Programme for research, technological development and demonstration under grant agreement number 607150.

## Author contributions

S.C-W. and P.R. contributed equally to the conception and design of the study, the data acquisition, and the analysis and interpretation of the data, for this reason P.R is listed as a joint first author; the manuscript was written by S.C-W. and P.R. was involved in manuscript revisions. S.K. contributed to the conception and design of the study, the data interpretation and the revision of the manuscript. D.G. contributed to data interpretation and manuscript revision. S.U. contributed to the data analysis and interpretation, as well as revisions of the manuscript. N.K. contributed to the data interpretation and revision of the manuscript. B.M. contributed to the statistical analysis and revision of the manuscript. D.H. contributed to the revision of the manuscript.

## Additional information

**Competing Interests:** The authors declare no competing interests.

