## [Peer Review File · Nature Communications]

Reviewers' comments:

Reviewer #1 (Remarks to the Author):

This paper addresses timely questions of how GHG emission from small rivers will change with changing climate, specifically, temperature. It concludes that sediment character is a key factor affecting the production of greenhouse gases, and the dominant gases released. There is a large amount of scientific interest in the release of CO₂ and CH₄ from inland waters, and areas of this paper touch on key controls, including sediment carbon, temperature, and substrate side.

I find the experimental approach, and some of the findings interesting -- though the paper as a whole is not well written, making key areas difficult to follow. Methodologically, I have major concerns re: whether the methods support the strength of conclusions that are presented. I also suggest important details on some of the methods would need to be added.

My most substantive concern is that ebullition is the dominant drive of CH₄ release, and CH₄ is the dominant gas. Temperature is often reported to impact CH₄ production -- but there are myriad factors that must be considered when considering ebullition rates. Sediment carbon, and particle size are two of those considerations. Given ebullition is the dominant pathway, and CH₄ emitted via ebullition is not subject to substantive oxidation -- then the attempt to scale this work to understand future climate is fraught with uncertainty.

The resazurin-RRU methodology is not explained in the paper, hence the paper becomes difficult to follow without pulling out the literature -- and this rationale must be articulated in a few sentences. Other areas where the rationale needs to be further explained are around line 74 -- comments re: insufficient substrate, with citations on bioirrigation. Here, the links / specificity of results need to be more clearly articulated. Insufficient substrate may be at play -- but how do the experimental conditions (which are artificial) affect the conclusions, and hence, how applicable are they to the system itself?

Line 96 discusses geological origin-- but a more proximate explanation (what is it about geologic origin) that matters -- and -- how generalizable are your insights beyond the direct study area?

In some areas, the text needs to be more specific -- e.g., line 132 the Q10 values...do not specify which process -- so the text can be revised to be easier to follow.

The CH₄:CO₂ ratios -- again -- could be interesting -- but -- really, the lack of inclusion of ebullition will drastically alter in-field ratios. Dissolved ratios may be impacted, but this strikes me as an area of limited interest.

Sediment surface area affects porosity, which affect the diffusion of oxygen into the sediments -- how will this affect results in your experimental systems?

The assumptions re: streambed warming and effects on CO₂ efflux to me are overstated -- (line 271) -- perhaps this is true in the simplified system -- but how might other processes affect efflux? primary producers as one consideration, changing pH as another.

I strongly disagree with the statement on line 276 -- as I don't believe you can directly scale sediment incubations to understanding emissions from surface waters.

Line 281 strikes me as self evident -- we know (and have for a long time) that sediments are the major site of photosynthesis and respiration in shallow sites -- so of course we need to consider sediment respiration. Perhaps this is just a fine point of phrasing -- but again, it tends to overstate significance of the work.

Line 305-- how long were sediments stored before experimentation.

Line 310 -- why a different water source? why not filter to remove microbes from natural water? The lack of any ions in ultrapure water is sure to impact the microbes.

Line 314-- I'm not following the replication design in this section.

Reviewer #2 (Remarks to the Author):

Review of the manuscript NCOMMS-17-16889-T entitled “Thermal sensitivity of CO₂ and CH₄ emissions varies with streambed sediment properties”:

The objective of this paper is to assess the impact of streambed sediments' temperature on production of greenhouse gases and microbial metabolic activity. The authors incubated sediments from different geologic origins (chalk vs. sandstone streams) and with different organic matter content and substrata grain at five different temperatures (5, 9, 15, 21, and 26°C) for five hours and measured change in CO₂, CH₄, and resazurin-resorufin (tracer system used as a proxy for microbial metabolic activity). They also monitored changes in dissolved oxygen in the water column and the headspace.

The authors claims include that microbial metabolic activity increased with temperature across all substrates (L57) but that, in contrast to what has been reported in the literature, the increase wasn't linear (L64-65). Maximum increases in CO₂ production between 5°C and 26 °C were observed for fine sediments and CH₄ production rates did not increase with temperature across all substrates (L155).

The paper presents an interesting and valuable dataset and tackles a highly relevant question, however I have a few concerns that prevent me to recommend it for publication as it is, especially in relation to the data analysis and interpretation of the results. The paper could benefit from a more in deep discussion by including the possible causes of the observed results and the expected relationships between CO₂, CH₄ and Rru production rates and C10s. For instance, explain how CH₄, CO₂ and Rru production can happen simultaneously or how come there is a negative production of CO₂ under aerobic conditions in the, presumably, absence of primary producers. Also, Raz and Rru are known to sorb to sediments –how sorption might have affected the results especially in relation to the fine sediments and chalk vs. sandstone origins? Finally, I'm not a statistics expert but I thought the statistical approach to be rather odd – why not analyze the dataset as a unity instead of looking to pair to pair relationships? Aren't we increasing the probability to commit type I error and also missing the possible interactions between temperature and organic matter content? I would recommend a second opinion from a statistics expert.

Specific comments:

Microbial metabolic activity - Please clarify the units reported for metabolic activity -are they ng of Rru for every µg of Raz added to the bottle or for every µg of Raz recovered at the same time than Rru?

What were the ranges of observed Raz and Rru concentrations and how were they related to the standards? Please clarify if all samples were above the detection limit.

L60 “MMA in Chalk_{fine} increased by 1259%” - Is this number realistic or a reflection of the low activity at 5°C? What are the confidence intervals?

L63 “MMA was higher at Chalk_{fine} than Sandstone_{fine} despite the same o.m. content” Do you have any hypothesis that could explain it? Sorption artifact?

L70 “water column oxygen concentrations” – Is water column DO representative of sediment DO? How deep were the sediments? I've observed changes in DO in 5 cm depth after a few hours of incubation in mesocosms.

Temperature coefficient Q10 – please make sure to note in the text if you are referring to Q10_{MMA}, Q10_{CO2} or Q10_{CH4} (e.g. L132, L213, etc.).

Include a brief description of which temperatures were used to calculate Q10 in the methods and in table S2.

L225 – The sentence is long and confusing, I would replace it by 22 nmols CH₄ g⁻¹ hr⁻¹ at 3°C and 80 nmols CH₄ g⁻¹ hr⁻¹ at 22°C. This applies in general every time that there is a “respectively” in the sentence (e.g. L126, L135, L228, etc.)

L213 “Q10 values generally ranged between 0 and 4 – half of them were higher than 4...”

Determination of organic matter content - Include the sediment subsample size that was combusted –See Heiri, O., A. F. Lotter and G. Lemcke (2001). "Loss on ignition as a method for estimating organic and carbonate content in sediments: reproducibility and comparability of results." *Journal of Paleolimnology* 25(1): 101-110, and Santisteban, J. I., R. Mediavilla, E. López-Pamo, C. J. Dabrio, M. B. R. Zapata, M. J. G. García, S. Castaño and P. E. Martínez-Alfaro (2004). "Loss on ignition: a qualitative or quantitative method for organic matter and carbonate mineral content in sediments?" *Journal of Paleolimnology* 32(3): 287-299.

Figures – Remove regression lines when they are not statistically significant. Including them can be deceiving.

Reviewer #3 (Remarks to the Author):

The manuscript entitled 'Thermal sensitivity of CO₂ and CH₄ emissions varies with streambed sediment properties' claims the importance of GHG emissions from streambed sediments to the overall C emissions. The authors used microcosms to test for the emission of GHG from streambed sediments of different properties (OM content, grain size, geology) at 5 different temperatures. Results of the microcosm experiment clearly demonstrate that temperature sensitivity of CO₂ and CH₄ emissions were highest in fine sediments from chalk geology having a high OM content. The temperature response is non-linear with a threshold between 15°C and 21°C. The results demonstrated in the manuscript are novel and important to better understand spatial and temporal heterogeneity in GHG emissions from stream ecosystems, since the carbon turnover differs with sediment property. In my opinion, the paper is important for other scientist working in the field of GHG emissions and global climate change. For public regulatory authorities the knowledge would be important to use these data on GHG production and its spatial variability within streambed sediments for better understanding and upscaling the effects on a global scale under climate change and groundwater abstraction scenarios. Therefore, I highly recommend publication of the paper in Nature Communications.

However, I have four suggestions to improve the quality of the manuscript.

Firstly, the authors underpin their results with a lot of numbers (mainly ranges) in the 'Results and Discussion' section, which makes this section very hard to read. I recommend reducing numbers and naming only those that are very important for the reader. Trends and size of results can easily be found in the Figures.

Secondly, the methods description is sound and the experiment easy to repeat following the description. I would like to see a description of the calculation and meaning of the Q10 values.

Thirdly, I recommend improving the statistical analyses. The setup of the experiment indicates a clear 2-factorial design with temperature and sediment property. In case, as I would suggest, applying a 2-factorial analysis of variance the residuals should show normal distribution and homogeneity of variance. This type of analysis gives information on the effect of temperature, sediment substrate property and their interaction on microbial activity and GHG emission. The data seem to result in a significant interaction term, showing that temperature is a driver of GHG emission in some sediments with a distinct substrate property but not in others.

And finally, for the illustration of data in graphs I recommend using points with standard deviation as in Fig. 3. To use only three data points (number of replicates in experiment) to create a box plot is to my knowledge not appropriate.

1. Reviewer #1 (Remarks to the Author):

1.1 This paper addresses timely questions of how GHG emission from small rivers will change with changing climate, specifically, temperature. It concludes that sediment character is a key factor affecting the production of greenhouse gases, and the dominant gases released. There is a large amount of scientific interest in the release of CO₂ and CH₄ from inland waters, and areas of this paper touch on key controls, including sediment carbon, temperature, and substrate side. I find the experimental approach, and some of the findings interesting -- though the paper as a whole is not well written, making key areas difficult to follow.

We followed the Reviewer's advice to improve the results and discussion section by reducing the many values reported in the text that made this difficult to follow. The recommended brevity in highlighting key points with quantitative data only in the main text was adopted in response to the Reviewer's recommendation. The majority of detailed values have now been removed from the text, and the detailed data are ready available in the presented tables and figures.

1.2 Methodologically, I have major concerns re: whether the methods support the strength of conclusions that are presented. I also suggest important details on some of the methods would need to be added.

My most substantive concern is that ebullition is the dominant drive of CH₄ release, and CH₄ is the dominant gas. Temperature is often reported to impact CH₄ production -- but there are myriad factors that must be considered when considering ebullition rates. Sediment carbon, and particle size are two of those considerations. Given ebullition is the dominant pathway, and CH₄ emitted via ebullition is not subject to substantive oxidation -- then the attempt to scale this work to understand future climate is fraught with uncertainty.

We fully agree that ebullition is an important transport process for CH₄ release in saturated wetland/peatland and river systems. Therefore, we designed our experimental approach to directly measure bulk fluxes of CH₄ due to background diffusional and episodic ebullition processes in enclosed jars. The jars in our experiment were incubated with the sediment and water column enclosed, and after 5 hours gas samples were taken from the headspace, where CH₄ had accumulated via both transport pathways. We recognise that this was not made explicitly clear within the original manuscript. Within the revised manuscript we have, therefore, clarified in the text that our incubation experiments include both ebullitive and diffusive fluxes.

"GHG samples were collected from the headspace of the incubation jars, and so fluxes measured here include both diffusive and ebullitive pathways." (Line 58 – all line numbers refer to those in the clean manuscript)

We acknowledge that our experiments have altered the structure of the sediment from natural conditions, which would have affected bubble entrapment. This would, however, have made sediment sites more uniform and so our data represents a conservative estimate of sediment CH₄ emissions as a function of substrate quality and temperature across the different geologies. We now include in the revised version of the paper a discussion of how ebullition increases in fine and organic matter-rich sediments. We state in the CH₄ discussion:

"Ebullition may be responsible for this observation as it is related to finer sediments⁴¹, explaining high production rates in Chalk_{fine} and Sandstone_{fine}, and OM-rich sediments^{41,42}, resulting in elevated production rates in the three highest OM content sediments investigated here." (Line 179)

1.3 The resazurin-RRU methodology is not explained in the paper, hence the paper becomes difficult to follow without pulling out the literature -- and this rationale must be articulated in a few sentences.

Following the suggestion of the Reviewer, we have amended the manuscript and added a substantially-enhanced, comprehensive description of the resazurin (Raz)-resorufin (Rru) system into the methodology of the revised manuscript.

"2.4 Aerobic Microbial Metabolic Activity (MMA)- Resazurin and Resorufin Concentration

The Raz-Rru system has been developed as a reactive tracer due to the utilisation of Raz as an electron acceptor in aerobic respiration, resulting in the irreversible conversion of Raz to Rru^{56,57}. Rru production can, therefore, be used as a proxy for aerobic microbial metabolic activity^{56,57}, Rru production is usually normalised by the amount of Raz detected in samples, to account for any losses of the tracer from the system. As we performed experiments in closed jars, we only measured Rru in the samples.” (Line 405)

1.4 Other areas where the rationale needs to be further explained are around line 74 -- comments re: insufficient substrate, with citations on bioirrigation. Here, the links / specificity of results need to be more clearly articulated. Insufficient substrate may be at play -- but how do the experimental conditions (which are artificial) affect the conclusions, and hence, how applicable are they to the system itself?

We would like to highlight that this was a short-term incubation experiment in which the total carbon produced was small (maximum of 849 ppm) relative to the amount of organic matter (minimum of 6670 ppm) present in the jars, therefore, suggesting that there was sufficient substrate within the jars throughout the experiment. Given this fact, it is likely that the fluxes were more greatly influenced by the quality of the organic matter, and thus, the influence of organic matter quantity is likely to have been negligible, if any. In line with the Reviewer’s comment, therefore, we removed the suggestion of substrate limitation in the manuscript and also, therefore, of metabolic suppression due to insufficient substrate from the manuscript.

We do agree with the Reviewer that conditions in an incubation experiment differ from that within the natural system. Our experimental design was optimized to approximate natural conditions, while controlling key proximal controllers of greenhouse gas fluxes, to explain the direct impact of changing conditions, such as temperature, on greenhouse gas emission potential. Since the study was designed to elucidate the role of changing temperature under controlled experimental conditions, our results point to a potential shift in the greenhouse gas balance of rivers under climate change. However, these results cannot be extrapolated to in situ conditions to estimate bulk greenhouse gas fluxes. Keeping in view the concerns of the Reviewer here, we have highlighted and elaborated the scope of the findings within the means of the data. Please see points 1.8 and 1.9 below for more detail.

Moreover, there will be some discrepancies in translating the lab-based incubation results to future field conditions. The focus of our experiment, however, was on highlighting the role of changing environmental conditions on the potential strength and/or direction of greenhouse gas fluxes from sediments with different organic matter contents, rather than quantifying the long-term extent of in situ fluxes.

1.5 Line 96 discusses geological origin-- but a more proximate explanation (what is it about geologic origin) that matters -- and -- how generalizable are your insights beyond the direct study area?

Different geologies have varying biogeochemical properties and are, therefore, expected to have different greenhouse gas production rates. Hence, geology was included as a potential controlling factor in the experimental design. Sandstone and carbonate geologies were investigated in this study as they cover the range of dominant aquifers for the whole of Britain and also globally, allowing any conclusions from the study to be more generalisable beyond the immediate study area. We acknowledge that we did not provide an explanation of this in the manuscript, and so have done so in the revised manuscript.

The rationale behind choosing two different geologies, has been added to the revised manuscript as follows:

“Geological origin was investigated as different geologies have varying biogeochemical properties and, therefore, are expected to have different GHG production rates.” (Line 47)

The reason for specifically choosing sandstone and carbonate has been added to the revised manuscript as follows:

“Sandstone and carbonate were investigated as these are the dominant aquifer materials of the globe, therefore, allowing any conclusions to be more generalisable.” (Line 53)

1.6 In some areas, the text needs to be more specific -- e.g., line 132 the Q10 values...do not specify which process -- so the text can be revised to be easier to follow.

The references to Q_{10} within the manuscript have been replaced as suggested by the Reviewer with the following notation, which indicates whether the process in question is MMA, CO₂ or CH₄ production: Q_{10MMA} , Q_{10CO_2} and Q_{10CH_4} .

1.7 The CH₄:CO₂ ratios -- again -- could be interesting -- but -- really, the lack of inclusion of ebullition will drastically alter in-field ratios. Dissolved ratios may be impacted, but this strikes me as an area of limited interest. Sediment surface area affects porosity, which affect the diffusion of oxygen into the sediments -- how will this affect results in your experimental systems?

We would like to clarify that ebullition is included in our experiments, as discussed above, and so the discussion of CH₄:CO₂ ratios is valuable and we believe of interest to readers.

The ratio of CH₄:CO₂ is affected by the concentration of dissolved O₂ in the sediments and this will be influenced by sediment surface area and porosity as indicated by the Reviewer. The sediment was homogenised manually prior to the incubation experiments to elucidate and isolate the influence of temperature on aerobic and anaerobic microbial metabolism, therefore, the incubated sediments represent potential rather than in situ process rates. We tried to emulate the real world as closely as possible by manually swirling sediment slurries after the addition of the ultrapure water, to allow for re-sorting of sediments with heavier particles at the base and finer particles settled on top, to mimic natural sediment sorting and deposition conditions. Thus, we believe that our incubated sediments approximated natural conditions to enable the measurement of potential changes in process rates that might happen in nature under future temperature change scenarios. For example, once incubated, the system was kept undisturbed to avoid any impact either on dissolution of headspace O₂ into the water and sediments, and the ebullition-based fluxes of greenhouse gases, particularly CH₄. This allowed for comparing fluxes from sediments of different texture and organic matter content across the different geologies under a constant sediment surface area. Thus, differences in sediment porosity constraints on oxygen diffusion were minimized to avoid experimental anomalies as pointed out by the Reviewer. We amended the manuscript accordingly to clarify these points, particularly in the methodology.

"To enable the results from these incubation experiments to reflect in situ processes, real world conditions were emulated as closely as possible by manually swirling sediment slurries after the addition of the ultrapure water, to allow for re-sorting of sediments with heavier particles at the base and finer particles settled on the top, to mimic natural sediment sorting and deposition conditions. Additionally, once incubated, the disturbance within the system was minimised to avoid any impact either on dissolution of headspace O₂ into the water and sediments, and the ebullition-based fluxes of GHG, particularly CH₄. This allowed for comparing fluxes from sediments of different texture and organic matter content, across the different geologies, under a constant sediment surface area. Thus, differences in sediment porosity constraints on oxygen diffusion were minimized to avoid experimental anomalies, and subsequent effects on production rates and CO₂:CH₄ ratios." (Line 357)

1.8 The assumptions re: streambed warming and effects on CO₂ efflux to me are overstated -- (line 271) -- perhaps this is true in the simplified system -- but how might other processes affect efflux? primary producers as one consideration, changing pH as another.

While we appreciate that our one component analysis of a complex, real-world system does not capture all influences on sediment greenhouse gas production and surface water efflux, it does allow for a systematic investigation of the proximal drivers of greenhouse gas fluxes, including differences in organic matter substrate content, degree of anoxia and temperature. Knowing the interactive role of distal controllers of greenhouse gas fluxes, such as bioturbation, activity of primary producers and eventual net primary productivity, and pH on influencing the proximal controllers would have been useful, however, this was beyond the scope of the current research. We acknowledge the concern of the Reviewer here; however, our results integrate the impact of differences in distal controllers of greenhouse gas fluxes broadly through our focus on the proximal controllers in the laboratory incubation. Thus, our statements about the impacts of streambed warming can be seen as rather conservative and reflect more of the direct impact on greenhouse gas fluxes. We amended the manuscript accordingly as suggested by the Reviewer.

“Comparisons between incubation experiment results and global observations allow the quantification of the effect of streambed GHG production, in relation to increasing temperature, on GHG efflux from streams and rivers. Incubation experiments and global observations represent different scales and natural conditions; therefore, these increases in effluxes represent an estimate of the influence of streambed GHG production at different temperatures and some caution needs to be exercised when predicting field fluxes under future climate change.” (Line 300)

1.9 I strongly disagree with the statement on line 276 -- as I don't believe you can directly scale sediment incubations to understanding emissions from surface waters.

We considered the Reviewer's concerns and have modified the text in the revised manuscript to highlight the conceptual and technical limitations of upscaling incubation experiments to surface water emissions. We emphasise that our quantifications represent first approximations to allow sediment greenhouse gas production potential to be incorporated into surface water observations:

“While the issues with upscaling incubation experiments to surface water emissions are appreciated, when used as an approximation (as above)” (Line 307)

The 'as above' in this text refers to that text in point 1.8.

1.10 Line 281 strikes me as self evident -- we know (and have for a long time) that sediments are the major site of photosynthesis and respiration in shallow sites -- so of course we need to consider sediment respiration. Perhaps this is just a fine point of phrasing -- but again, it tends to overstate significance of the work.

Although we agree with the Reviewer that we have known for a long time that sediments are important in stream respiration, this has been largely overlooked when considering stream greenhouse gas production and carbon efflux. Our statement, therefore, addresses not the importance of streambed respiration, but the importance of considering it in the assessment of greenhouse gas emission scenarios, and we have amended the revised manuscript to clarify that this point has been largely overlooked previously.

“Our results indicate the importance of considering streambed respiration, which has been largely unaccounted for in C efflux despite knowledge of its importance^{8,54}, and subsequent CO₂ production,” (Line 313)

1.11 Line 305-- how long were sediments stored before experimentation.

The sediments were stored for 3 weeks before the incubation experiments began. Each incubation temperature treatment was performed in a separate week, over a period of 7 weeks. Therefore, the sediments were stored for a maximum of 9 weeks prior to the final temperature treatment. To minimise any effect due to the length of sediment storage, the order in which the temperature treatments were performed were randomised, instead of performing the temperature treatments in order from lowest to highest temperature. We thank the Reviewer for their comment, which made us realise we did not include this in the manuscript, and have now clarified this in the methodology of the revised manuscript.

“The sediment was then stored air tight, in the dark, at 4±1°C, for 9 weeks between collection and the beginning of the last temperature incubation. To minimise any potential effects of sediment storage the order of the temperature treatments was randomised, rather than performed in sequence e.g. from the lowest to highest temperature.” (Line 339)

1.12 Line 310 -- why a different water source? why not filter to remove microbes from natural water? The lack of any ions in ultrapure water is sure to impact the microbes.

The experiment was designed so that sediment greenhouse gas production could be investigated independently of influences from stream water solutes and microbes, to achieve this, therefore, it was necessary to use ultrapure water instead of stream water to prevent any variability in solutes, dissolved organic carbon or microbes in the stream water from affecting the results.

Additionally, the application of ultrapure water improves the repeatability of this well-designed factorial experiment. Incubations were performed over a 7-week period making it impossible to perform the different temperature incubations with a consistent stream water

source. If the stream water was collected prior to the incubations and stored, then its composition would change over time; alternatively, if new stream water was collected weekly for each set of incubations, the conditions in the stream would likely have changed and so the composition would again be inconsistent throughout the incubation experiments. These inconsistent stream water compositions could have significant impacts on the incubation experiments.

The following explanation of why ultrapure water was used instead of stream water, has been added to the methodology:

“Ultrapure water was used to allow sediment processes to be investigated independently of stream water solutes and microbes.” (Line 347)

1.13 Line 314-- I'm not following the replication design in this section.

We appreciate that the explanation of the experimental design was difficult to follow. This section has been re-worded in the revised manuscript to clarify the experimental and replication design.

“The 3 substrate types from the 2 geological origins, as well as the control experiments, gave a total of 7 substrates, which were ran in triplicate resulting in 21 incubation jars per temperature treatment (Figure 2). Once the samples were added the jars were swirled and placed, with lids ajar, into an incubation oven at treatment temperature for 3 days prior to the beginning of the incubation time period. The incubations were performed for 5 hours at 5, 9, 15, 21 and 26°C. Sampling occurred at 0 and 5 hours as described below.” (Line 349)

2. Reviewer 2

Review of the manuscript NCOMMS-17-16889-T entitled “Thermal sensitivity of CO₂ and CH₄ emissions varies with streambed sediment properties”: The objective of this paper is to assess the impact of streambed sediments' temperature on production of greenhouse gases and microbial metabolic activity. The authors incubated sediments from different geologic origins (chalk vs. sandstone streams) and with different organic matter content and substrata grain at five different temperatures (5, 9, 15, 21, and 26°C) for five hours and measured change in CO₂, CH₄, and resazurin-resorufin (tracer system used as a proxy for microbial metabolic activity). They also monitored changes in dissolved oxygen in the water column and the headspace. The authors claims include that microbial metabolic activity increased with temperature across all substrates (L57) but that, in contrast to what has been reported in the literature, the increase wasn't linear (L64-65). Maximum increases in CO₂ production between 5oC and 26 oC were observed for fine sediments and CH₄ production rates did not increase with temperature across all substrates (L155). The paper presents an interesting and valuable dataset and tackles a highly relevant question, however I have a few concerns that prevent me to recommend it for publication as it is, especially in relation to the data analysis and interpretation of the results.

2.1 The paper could benefit from a more in deep discussion; by including the possible causes of the observed results and the expected relationships between CO₂, CH₄ and Rru production rates and C10s. For instance, explain how CH₄, CO₂ and Rru production can happen simultaneously or how come there is a negative production of CO₂ under aerobic conditions in the, presumably, absence of primary producers.

We thank the Reviewer for their helpful comments on where the manuscript could benefit from further discussion, and include their suggestions within the revised manuscript.

Resorufin (Rru) production has been developed as a proxy for quantifying aerobic microbial respiration, implying that substantial CH₄ and Rru should not be produced simultaneously within the incubation jars. However, as the sediment within the jars may have been heterogenous, it is possible that areas of oxic and anoxic sediments, and therefore, both aerobic and anaerobic respiration, occurred within the jars at the same time. This was observed during incubation of aerated soil columns by Parkin, 1987. This would lead to the production of Rru and CO₂ from the oxic areas, and the production of CH₄ and CO₂ from the anoxic areas. A discussion of this and how Rru, CO₂ and CH₄ could be produced simultaneously in the jars has been added to the revised manuscript.

“Although Rru production has been developed as a proxy for aerobic microbial respiration, it is possible that Rru is produced in the jar alongside methanogenesis, due to oxic and anoxic sediments existing simultaneously⁴⁵. This explains, therefore, the observations in some sediment types of simultaneous Rru, CO₂ and CH₄ production, especially at higher temperatures.” (Line 259).

The negative production of CO₂ under aerobic conditions, observed in Chalk_{coarse} at 5 and 9°C, could be caused by two processes. Firstly, as carbonate systems are sinks for CO₂ at high pH, some of the CO₂ produced may have remained dissolved in the water column as carbonate, rather than diffusing into the headspace. Negative CO₂ production due to this process has been observed previously and occurred above a pH of 8.5 (Therrien et al., 2005). The average pH in the Chalk_{coarse} sediments at 5 and 9°C was 8.3±0.1, close to this threshold of 8.5, and so it is possible that the negative production observed here was due to CO₂ remaining in the water column as carbonate. Secondly, the production was calculated as the concentration of CO₂ in the headspace at t=5 hours minus the CO₂ concentration in the headspace at t=0 hours. As Chalk_{coarse} has such low CO₂ concentrations at higher temperatures, it is possible that the negative production seen at 5 and 9°C is from headspace CO₂ from the atmosphere dissolving into the water column, and so reducing the CO₂ concentration in the headspace over time. This has been added into the revised manuscript:

“Somewhat surprisingly, in Chalk_{coarse} at 5 and 9°C, negative production of CO₂ was observed under aerobic conditions, which may be due to two processes. Firstly, negative production of CO₂ may occur at a high pH due to the aqueous carbonate system acting as a CO₂ sink. This has been observed previously at pH greater than 8.5³⁸, and the average pH of Chalk_{coarse} at 5 and 9°C was 8.3±0.1, close to this threshold. It is possible, therefore, that the CO₂ produced here did not diffuse into the headspace, but remained in solution as carbonate. Secondly, the production rates were calculated from the difference in concentration between 0 and 5 hours. As Chalk_{coarse} had such low headspace CO₂ concentrations, it is possible that CO₂ from the atmosphere dissolved into the water column at these low temperatures, reducing the CO₂ concentration in the headspace over time.” (Line 125)

Negative production also occurred in Tern medium sediments at 21°C in the figures sent in the original submission, however, we looked into this and there was an erroneous data point for a missing gas sample at 21°C in Tern medium, this has now been removed and the CO₂ production is now positive, as expected.

The conversion of resazurin (Raz) to Rru is irreversible so is not affected by primary producers, this has been clarified in the methods with a more detailed description of the Raz-Rru system, which now details that the conversion is irreversible.

“The Raz-Rru system has been developed as a reactive tracer due to the utilisation of Raz as an electron acceptor in aerobic respiration, resulting in the irreversible conversion of Raz to Rru^{56,57}.” (Line 406)

2.2 Also, Raz and Rru are known to sorb to sediments –how sorption might have affected the results especially in relation to the fine sediments and chalk vs. sandstone origins?

The sorption may be expected to differ if the organic matter contents of the sediments were very different, however, our sediments were all of relatively low organic matter content. Lemke et al., 2014 conducted in depth laboratory experiments, based on observations by others of sorption and a lack of complete mass recovery during tracer experiments. They compared Raz and Rru sorption in sediments from two streams; one dominated by sandstone and marlstone, and the other by limestone. The sediments also had differences in organic carbon content between the rivers, of 0.73% in the limestone and 0.13% in the sandstone and marlstone. They found no correlation between sorption and organic carbon content or sediment type, therefore, we expect little impact. As there is some sorption of the Rru to the sediment, we may have underestimated metabolism, however, this would be underestimated throughout the sediment types. Additionally, as the Reviewer suggests, there may be an increase in Rru sorption in the fine sediment, due to an increase in sediment surface area. These fine sediments, however, are where we see the highest Rru production, and so although we may have greater underestimation of metabolism in the fine sediments, this would not affect our conclusions that metabolism was greatest in fine sediments.

2.3 Finally, I'm not a statistics expert but I thought the statistical approach to be rather odd – why not analyze the dataset as a unity instead of looking to pair to pair relationships? Aren't we

increasing the probability to commit type I error and also missing the possible interactions between temperature and organic matter content? I would recommend a second opinion from a statistics expert.

We acknowledge the Reviewer's concerns regarding the group-wise comparisons performed for the statistical analysis and the high probability of Type-1 errors being introduced during this approach. We have taken the Reviewer's suggestion and enlisted the help of a statistical expert. The revised statistical analysis is based on residual maximum likelihood to estimate parameters for an ANOVA-type model, which was used to test whether our hypotheses were significant using a Wald's test, as demonstrated by Lark and Cullis 2004.

We have now added a statistical analysis section into the methodology, with tables of the results added to the supplementary information (Table S-1, Table S-2 and Table S-3). Additionally, we have added into the relevant areas that results were statistically significant and included p-values.

"2.7 Statistical Inference

We base our inference regarding the relationships between temperature, sediment type and GHG emissions upon the linear model:

$$(2) \quad y(s, t) = \sum_{i=0}^6 \{I_i(s)\alpha_i + I_i(s)\beta_i t\} + \varepsilon,$$

where $y(s, t)$ is the measured emission from sediment class s at temperature t , $I_i(s)$ is an indicator function that is equal to one when $i = s$ and is zero otherwise, the α_i and β_i are model coefficients for the intercept and gradient terms and each ε is independent and realized from a Gaussian distribution with zero mean and constant variance. Thus, each response curve is represented by a linear function. The seven sediment classes are: 0 – control, 1 – chalk fine, 2 – chalk medium, 3 – chalk coarse, 4 – sandstone fine, 5 – sandstone medium and 6 – sandstone coarse.

We estimate the model coefficients $\mu = [\alpha_0, \dots, \alpha_6, \beta_1, \dots, \beta_6]^T$ for each GHG variable (where T denotes the transpose) by maximum likelihood. Where the residuals that result are inconsistent with the Gaussian assumption, a shift and a log transform are applied to the data and the model is re-estimated until approximately Gaussian residuals are achieved.

We initially apply ANOVA tests to the null hypotheses that $\alpha_0 = \alpha_1 = \dots = \alpha_6$ and $\beta_0 = \beta_1 = \dots = \beta_6$. If these hypotheses can be rejected at the 0.05 level we then explore the factors causing variation in the α_i and/or β_i . We control the Type-1 errors across the experiment by testing a series of planned orthogonal contrasts. For the intersect terms these contrasts are:

C1: $\alpha_0 - \frac{1}{6} \sum_{i=1}^6 \alpha_i$, i.e. comparing the intercept of the control to the average of the six sediment types;

C2: $\frac{1}{3} \sum_{i=1}^3 \alpha_i - \frac{1}{3} \sum_{i=4}^6 \alpha_i$, i.e. comparing the intercepts for chalk to those for sandstone;

C3: $\alpha_1 - \frac{\alpha_2 + \alpha_3}{2}$, i.e. comparing the intercept for chalk fine to the average of that for the other chalk classes;

C4: $\alpha_4 - \frac{\alpha_5 + \alpha_6}{2}$, i.e. comparing the intercept for sandstone fine to the average of that for the other sandstone classes;

C5: $\alpha_0 - \frac{\alpha_1 + \alpha_4}{2}$, i.e. comparing the intercept for the control to the average of those for fine sediment classes.

If appropriate, the same comparisons were applied to the β_i gradient coefficients. The probability of achieving the estimated value of each contrast if the true value is zero, was then calculated by a Wald test following the methodology described by Lark and Cullis (2004).

If either of the initial ANOVA hypotheses cannot be rejected then the corresponding α_i or β_i in the model are replaced by a single constant coefficient and the remaining parameters are re-estimated before the above contrasts are tested." (Line 452)

Table S-1. Estimated value of each intersect and gradient contrast and probability of these estimates being achieved under the null model $C_i=0$ for $\log(Rru + 0.1)$. Significant p-values are in bold.

Contrast	Intersect: α_i		Gradient: β_i	
	Estimated value	p	Estimated value	p
ANOVA		<0.01		<0.01
C1	-0.13	0.82	-0.16	<0.01
C2	1.16	0.14	0.01	0.64
C3	3.28	<0.01	-0.04	0.35
C4	2.11	<0.01	0	0.95
C5	-1.57	0.11	-0.29	<0.01

Table S-2. Estimated value of each intersect and gradient contrast and probability of these estimates being achieved under the null model $C_i=0$ for $\log(CO_2 + 0.1)$. Significant p-values are in bold.

Contrast	Intersect: α_i		Gradient: β_i	
	Estimated value	p	Estimated value	p
ANOVA		<0.01		<0.01
C1	-1.82	<0.01	-1.01	<0.01
C2	-6.01	<0.01	0.46	<0.01
C3	10.59	<0.01	1.47	<0.01
C4	12.93	<0.01	0.65	<0.01
C5	-15.79	<0.01	-3.57	<0.01

Table S-3. Estimated value of each intersect and gradient contrast and probability of these estimates being achieved under the null model $C_i=0$ for $\log(CH_4 + 0.01)$. Significant p-values are in bold.

Contrast	Intersect: α_i		Gradient: β_i	
	Estimated value	p	Estimated value	p
ANOVA		0.17		<0.01
C1	NA	NA	-0.08	<0.01
C2	NA	NA	0.06	<0.01
C3	NA	NA	0.16	<0.01
C4	NA	NA	0.14	<0.01
C5	NA	NA	-0.36	<0.01

Specific comments:

2.4 Microbial metabolic activity - Please clarify the units reported for metabolic activity -are they ng of Rru for every μg of Raz added to the bottle or for every μg of Raz recovered at the same time than Rru?

We thank the Reviewer for pointing out that the units of microbial metabolic activity presented in the manuscript were ambiguous. The units are ng of Rru per μg of Raz added to the jar at $t = 0$ hours, and this has now been clarified at the start of the results section for microbial metabolic activity.

“MMA results are presented as ng of Rru produced per μg of Raz added to the jar at time zero” (Line 64)

2.5 What were the ranges of observed Raz and Rru concentrations and how were they related to the standards? Please clarify if all samples were above the detection limit.

Reference to the measurement of resazurin (Raz) has been removed from the methodology, as here only resorufin (Rru) production is presented. The range of Rru concentrations measured during the experiment, as well as a comparison of these to the limit of detection and standards has been added to the methodology.

“The concentration of Rru in the measured samples ranged from 0.0 to 139.6 ppb, therefore, some samples were below the limit of detection of the fluorometers. The maximum hourly production rate (dependent on the amount of Raz added to the jar) yielded from a Rru concentration of 1 ppb was $1.7 \text{ ng Rru } \mu\text{g}^{-1} \text{ Raz hr}^{-1}$, and any samples below the LOD are presented here as actual values, accounting for some of the low MMA rates observed.

The design of the GGUN FL30 fluorometer allows a one-point calibration of 100 ppb Rru to be sufficient for these concentrations, here a two-point calibration was performed to improve data quality.” (Line 418)

2.6 L60 “MMA in Chalkfine increased by 1259%” - Is this number realistic or a reflection of the low activity at 5oC? What are the confidence intervals?

This number is realistic and is a reflection of the low activity at 5°C. Based on this comment we feel we presented the figure at too high precision and so this has been rounded in the revised manuscript to 1260%.

2.7 L63 “MMA was higher at Chalkfine than Sandstonefine despite the same o.m. content” Do you have any hypothesis that could explain it? Sorption artifact?

We hypothesise that this difference in MMA was due to a difference in carbon quality between the two sediment types as the carbon in the Chalk_{fine} sediment had a lower aromaticity than that in the Sandstone_{fine} sediment. Additionally, the CO₂ and CH₄ production was also greater in Chalk_{fine} than Sandstone_{fine}, indicating there is a difference in activity between these sediment types. We have added a sentence to the revised manuscript highlighting the difference in carbon quality between the two sediment types:

“We suggest that this is due to differences in the aromaticity of the carbon, which was 17.3% in Chalk_{fine} and 20.9% in Sandstone_{fine} sediments. Carbon in Chalk_{fine} had a lower aromaticity, hence, the carbon was more bioavailable, producing greater MMA” (Line 77)

We believe, therefore, that there was not a sorption artefact as suggested by the Reviewer, due to the reasons given above and that Lemke et al., 2014 found no correlation of sorption between sediments from limestone and sandstone rivers.

2.8 L70 “water column oxygen concentrations” – Is water column DO representative of sediment DO? How deep were the sediments? I’ve observed changes in DO in 5 cm depth after a few hours of incubation in mesocosms.

The sediment was a maximum depth of approximately 5 cm in the incubation jars, and samples were taken at 5 hours, as the Reviewer suggests, there may be some differences between water column and sediment dissolved oxygen, but we expect these to be minimal.

2.9 Temperature coefficient Q10 – please make sure to note in the text if you are referring to Q10MMA, Q10CO2 or Q10CH4 (e.g. L132, L213, etc.).

We thank the Reviewer for noting that it was unclear in the manuscript which process was being referred to when discussing Q₁₀ values. This has now been addressed in the text and the Reviewer's suggestion of nomenclature has been followed by using Q_{10MMA}, Q_{10CO2} and Q_{10CH4} to allow the process to be referred to more easily during the Q₁₀ discussions.

2.10 Include a brief description of which temperatures were used to calculate Q10 in the methods and in table S2.

We had not clearly included the temperatures used to calculate the Q₁₀ values and so these temperatures have been clarified in a new Q₁₀ section of the methodology in the revised manuscript, as well as in Table S2.

"Q₁₀ values were calculated between the different incubation temperatures, so that T₁ and T₂ were 5 and 9, 9 and 15, 15 and 21, and 21 and 26°C (see Table S-2)." (Line 448)

2.11 L225 – The sentence is long and confusing, I would replace it by 22 nmols CH4 g⁻¹ hr⁻¹ at 30°C and 80 nmols CH4 g⁻¹ hr⁻¹ at 22°C. This applies in general every time that there is a "respectively" in the sentence (e.g. L126, L135, L228, etc.)

We agree that the manuscript was difficult to follow at times due to too many quoted values and sentence structures. This sentence has now been modified based on the Reviewer's suggestion, as well as other relevant sentences within the manuscript.

2.12 L213 "Q10 values generally ranged between 0 and 4 – half of them were higher than 4...

The Reviewer is correct and half of the Q₁₀ values were greater than 4. This text has now been changed to reflect the note made by the Reviewer.

"48% of the Q_{10CH4} values ranged between 0.0 and 4.1" (Line 232)

2.13 Determination of organic matter content - Include the sediment subsample size that was combusted –See Heiri, O., A. F. Lotter and G. Lemcke (2001). "Loss on ignition as a method for estimating organic and carbonate content in sediments: reproducibility and comparability of results." Journal of Paleolimnology 25(1): 101-110, and Santisteban, J. I., R. Mediavilla, E. López-Pamo, C. J. Dabrio, M. B. R. Zapata, M. J. G. García, S. Castaño and P. E. Martínez-Alfaro (2004). "Loss on ignition: a qualitative or quantitative method for organic matter and carbonate mineral content in sediments?" Journal of Paleolimnology 32(3): 287-299.

The sediment subsample weight that was combusted has now been included in the methodology, as the Reviewer pointed out the importance of including this.

"The sample was dried at 105°C overnight and weighed, resulting in subsamples of 14.8 to 24.8 g of dry sediment." (Line 399)

We appreciate that the loss on ignition technique is not a quantitative measure of sediment carbon. We tried to directly measure the carbon content of the sediment, however, the content was too low for the method available using an organic elemental analyser. This is why we focussed here on organic matter content and do not mention total organic carbon itself. We used a range of sediment sample sizes, depending on how much sediment was left in the crucible after drying. While the sediment sample size may have had some effect on the loss on ignition data, our samples had relatively low organic matter contents and so there should be less of an issue with varying sample size as there is less organic matter to combust.

2.14 Figures – Remove regression lines when they are not statistically significant. Including them can be deceiving.

We agree that including regression lines which were not statistically significant was misleading. We have now removed all regression lines from all figures, because the regressions for each sediment type were not amongst the pre-planned comparisons considered in the experiment.

3. Reviewer #3 (Remarks to the Author):

The manuscript entitled 'Thermal sensitivity of CO₂ and CH₄ emissions varies with streambed sediment properties' claims the importance of GHG emissions from streambed sediments to the overall C emissions. The authors used microcosms to test for the emission of GHG from streambed sediments of different properties (OM content, grain size, geology) at 5 different temperatures. Results of the microcosm experiment clearly demonstrate that temperature sensitivity of CO₂ and CH₄ emissions were highest in fine sediments from chalk geology having a high OM content. The temperature response is non-linear with a threshold between 15°C and 21°C. The results demonstrated in the manuscript are novel and important to better understand spatial and temporal heterogeneity in GHG emissions from stream ecosystems, since the carbon turnover differs with sediment property. In my opinion, the paper is important for other scientist working in the field of GHG emissions and global climate change.

For public regulatory authorities the knowledge would be important to use these data on GHG production and its spatial variability within streambed sediments for better understanding and upscaling the effects on a global scale under climate change and groundwater abstraction scenarios. Therefore, I highly recommend publication of the paper in Nature Communications.

However, I have four suggestions to improve the quality of the manuscript.

3.1 Firstly, the authors underpin their results with a lot of numbers (mainly ranges) in the 'Results and Discussion' section, which makes this section very hard to read. I recommend reducing numbers and naming only those that are very important for the reader. Trends and size of results can easily be found in the Figures.

We agree with the Reviewer that the many values included as ranges in the results and discussion section made this difficult to follow. The majority of these values have now been removed from the text to make this section clearer, with only values necessary to highlight key points now referenced in the text.

3.2 Secondly, the methods description is sound and the experiment easy to repeat following the description. I would like to see a description of the calculation and meaning of the Q₁₀ values.

We had not included a detailed description of the Q₁₀ methodology, which is important for understanding the manuscript, and thank the Reviewer for highlighting this. In the revised manuscript a Q₁₀ section has now been included in the methodology.

“2.6 Temperature coefficient values (Q₁₀)

The temperature coefficient value (Q₁₀) quantifies the temperature dependence of a biological process, and is here used to investigate the biological processes of MMA, and CO₂ and CH₄ production. The Q₁₀ value of a process is calculated using equation 1⁵⁷.

$$Q_{10} = \left(\frac{\text{Process}_{T_2}}{\text{Process}_{T_1}} \right)^{\left(\frac{10}{T_2 - T_1} \right)}$$

Where Process is the biological process under consideration at T₁ and T₂, and T₁ and T₂ are the respective temperatures at which Process was measured. Although Q₁₀ values are typically calculated where T₁ and T₂ are 10°C apart, using the form of the Q₁₀ equation given above allows T₁ and T₂ to have different temperature intervals. Q₁₀ values were calculated between the different incubation temperatures, so that T₁ and T₂ were 5 and 9, 9 and 15, 15 and 21, and 21 and 26°C (see Table S-2).” (Line 440)

3.3 Thirdly, I recommend improving the statistical analyses. The setup of the experiment indicates a clear 2-factorial design with temperature and sediment property. In case, as I would suggest, applying a 2-factorial analysis of variance the residuals should show normal distribution and homogeneity of variance. This type of analysis gives information on the effect of temperature, sediment substrate property and their interaction on microbial activity and GHG emission. The data

seem to result in a significant interaction term, showing that temperature is a driver of GHG emission in some sediments with a distinct substrate property but not in others.

We have performed new statistical analysis (please see point 2.3 above) based on the comments of Reviewers 2 and 3, using residual maximum likelihood to fit an ANOVA-type model.

3.4 And finally, for the illustration of data in graphs I recommend using points with standard deviation as in Fig. 3. To use only three data points (number of replicates in experiment) to create a box plot is to my knowledge not appropriate.

We have followed the advice of the Reviewer and all box plots have now been redrawn as scatter plots as in Fig. 3.

References

Parkin, T.B. 1987. Soil microsites as a source of denitrification variability. Soil Science Society of America Journal. 51, 1194-1199

Therrien J., Tremblay A., Jacques R.B. (2005) CO₂ Emissions from Semi-Arid Reservoirs and Natural Aquatic Ecosystems. In: Tremblay A., Varfalvy L., Roehm C., Garneau M. (eds) Greenhouse Gas Emissions — Fluxes and Processes. Environmental Science. Springer, Berlin, Heidelberg

Lark, R. and Cullis, B. 2004. Model-base analysis using REML for inference from systematically sampled data on soil. European Journal of Soil Science. 55(4), 799-813

Lemke, D., Gonzalez-Pinzon, R., Liao, Z., Wohling, T., Osenbruck, K., Haggerty, R. and Cirpka, O. A. 2014. Sorption and transformation of the reactive tracers Raz and Rru in natural river sediments. Hydrology and Earth System Sciences. 18(8), 3151-3163

Reviewers' comments:

Reviewer #1 (Remarks to the Author):

My most significant concern is that the experiment was a 5h slurry incubation including sediments (stored up to 10 weeks), and ultrapure water sources. The microbial community starts to change in hours, so there may be no resemblance to the natural community over weeks. Likewise, the ultrapure water is likely to lead to some degree of osmotic shock – hence while the rationale to exclude the streamwater microbial community may have some merit (although I suspect the abundances in the water column would be much, much lower), a more robust experimental design would require the work be done more quickly, and using communities not subject to long and variable storage times. I am also concerned re: variable oxygen in the systems based on comments in the response to line 357 and in comment 2.8. In response 2.8, re: water column oxygen – if sediments were 5cm deep, how deep was water? This will certainly have a large impact on oxygen – unless the water column is very deep.

Additional concerns are -- the experiment includes ebullition – which in some ways is a benefit; however, once the CH₄ is in a bubble, the CH₄ will be subject to little or no oxidation, affecting CH₄:CO₂ ratios. That means the mix of processes and phases will make it difficult to understand what is going on, or how to extrapolate it.

More minor comments -- Geological origin as related to biogeochemical properties was noted – can you articulate the specific properties that are important here? Alternatively, just noting the geologic origin is adequate if you can't go further – as written line 47 (re: biogeochemical properties) seems self evident.

Sorption/desorption – The other reviewer noted issues re: mass recovery. Is the magnitude of potential impact expected to be on the order of 1% or less? If so, I think this is OK – but this methodological issue requires a more detailed discussion re: justifying why the mass recovery experiments aren't needed, and why what we know suggests the implication for the results are limited. I think this is likely the case, but this is a significant issue that merits further discussion.

Re: response to line 281, I still disagree. Much of the GHG literature discusses productivity gradients with respect to CO₂ and CH₄ flux. The actual text is something where a small change might make it more acceptable. E.g., just note the fact it is important to consider streambed respiration in CO₂ production – rather than highlighting a research gap that I'm not sure exists.

Reviewer #2 (Remarks to the Author):

Dear editor and authors,

I have now reviewed the resubmission of the manuscript NCOMMS-17-16889A and I believe that it has improved to the point to recommend for publication. The authors have satisfactorily addressed all my comments. I especially value the new statistical approach and the streamline of the results section. The readability of the manuscript has substantially increased as a result.

Best regards,

Reviewer #3 (Remarks to the Author):

In the revised version of the manuscript about green house gas emission from different sediments of small streams the authors improved the statistical methods, the figures and the reading flow of the results and discussion section as suggested. I have no more concerns and recommend publishing this interesting and timely study in Nature Communications. There are two minor changes in the Material section highlighted in the attached pdf file.

Author's response:

We thank the editor and reviewers for their insightful comments. Please find below our point-by-point responses to the Reviewer's comments and details of how Reviewer suggestions have been implemented in the revised manuscript. All line numbers refer to those within the manuscript containing track changes.

You will see from the comments that while they find your work of interest, Reviewer 1 continues to raise important points. We are interested in the possibility of publishing your study in Nature Communications, but would like to consider your response to these concerns in the form of a revised manuscript before we make a final decision on publication.

We therefore invite you to revise and resubmit your manuscript, taking into account the points raised. In this revised version, we urge to pay particular attention to Reviewer 1's concerns regarding the study's experimental setup and how it may impact its findings. We ask that you explicitly discuss the limitations of your approach in your revised manuscript. Please note that in order to consider a revised manuscript, all concerns must be addressed. Please highlight all changes in the manuscript text file.

We have addressed the Reviewer's comments, paying particular attention to Reviewer 1's concerns regarding the experimental setup, please see points 1.1 and 1.2 for further detail. We have addressed the Reviewer's specific points in detail and also outlined the limitations of the experiment in the method section, explicitly highlighting the fact that incubation experiments, such as these, determine potential rates of microbial metabolic activity and greenhouse gas production from streambed sediments:

“2.8 Experimental limitations

The results of incubation experiments such as the one conducted here allow for systematic analyses of potential rates of MMA and greenhouse gas production. Since such systematic analyses are always based on model systems, one to one comparisons with site-specific in-situ measurements under less controlled conditions can remain challenging. Instead, systematic incubation studies support generalisations of the dependence of microbial metabolic activity and greenhouse gas production to varying temperature, substrate type, OM content and geological origin. Despite the fact that, like all methods, laboratory incubation experiments have limitations, they provide powerful tools to systematically investigate the impact of individual drivers, and remain the best way to control experimental treatment conditions that, in contrast to site-specific field studies, warrant generalisations of results.

In this study, the duration of experimental treatments required that sediment be stored over the duration of the entire experiment and mixed with ultrapure water instead of river water to guarantee stationarity of chemical conditions in the applied water source. The risk of this affecting the experimental results, e.g. by causing some alteration of the microbial community over the period of sediment storage, has been minimal; however, the sediment was stored in cold and dark conditions and the order of temperature treatments were randomised to avoid any time-lapsed anomalies in the results as a function of temperature. The risk of critical dilution of pore-waters with

ultrapure water that would have the potential to affect microbial communities is considered minimal since sediment and pore-water solute concentrations, such as nutrients, in the investigated lowland streams are high, with the ultrapure water addition likely to cause only minimal effects to the microbial communities that tend to be associated with sediment particles and so likely only experienced some dilution of porewaters. This, combined with the large responses to temperature increases in the fine sediments that were observed in this study, indicate that the microbial communities were not greatly affected by osmotic shock or similar processes.” (Line 576).

We have also summarised and referred to these limitations within the results and discussion section as below:

“Laboratory incubation experiments, while allowing individual drivers to be investigated, may not compare one-to-one to in-situ conditions within the natural system where boundary conditions are less constrained (see section 2.8). As such, the results of this study provide potential rates of MMA and GHG production, which indicate the difference in magnitude of rates between sediments of varying substrate, OM content and geological origin, and differences in response to temperature due to these varying characteristics.” (Line 338).

Reviewers' comments:

Reviewer #1 (Remarks to the Author):

1.1 My most significant concern is that the experiment was a 5h slurry incubation including sediments (stored up to 10 weeks), and ultrapure water sources. The microbial community starts to change in hours, so there may be no resemblance to the natural community over weeks.

We agree with the Reviewer that care needs to be taken when performing incubation experiments to ensure that results from laboratory experiments are relevant to in-situ conditions. As outlined in detail in our added content on experimental design and limitations above, to minimise the impact on microbial communities, sediments in this experiment were stored cold and in the dark. Additionally, the order in which the temperature treatments were performed was randomised so that there was no correlation between temperature and storage time. The randomisation of temperature treatments is stated on line 412 in the manuscript.

We have, on the advice of the Reviewer and Editor, also included a new section specifically discussing the limitations in the applied methodology which discusses the potential issues associated with laboratory incubation experiments, and in particular sediment storage times within the revised manuscript:

“The risk of this affecting the experimental results, e.g. by causing some alteration of

the microbial community over the period of sediment storage, has been minimal; however, the sediment was stored in cold and dark conditions and the order of temperature treatments were randomised to avoid any time-lapsed anomalies in the results as a function of temperature.” (Line 589).

1.2 Likewise, the ultrapure water is likely to lead to some degree of osmotic shock – hence while the rationale to exclude the streamwater microbial community may have some merit

While we agree with the Reviewer that there could potentially be issues with ultrapure water additions if microbial communities were directly exposed to a porewater to ultrapure water substitution, we like to highlight that this is by no means the case in this study. Rather, microbial communities in contact with sediment and porewater will have experienced a porewater dilution, which given the high nutrient and carbon content in the typical lowland rivers investigated, will have been far outside any critical conditions for the microbes. We furthermore interpret the significant metabolic response of the healthy microbial community within the fine sediments that critical issues such as osmotic shock can be excluded for these experiments. The risk of osmotic shock was reduced as the ultrapure water was added to the sediment and so there was a dilution of the porewater rather than stressing of the microbial community by placing it into ultrapure water alone. Additionally, the majority of microbes are associated with sediment particles and would remain within the sediment, and the sediments were taken from nutrient heavy streams^{1,2}, further protecting them from osmotic shock.

We have discussed the limitations of using ultrapure water within the revised manuscript:

“The risk of critical dilution of pore-waters with ultrapure water that would have the potential to affect microbial communities is considered minimal since sediment and pore-water solute concentrations, such as nutrients, in the investigated lowland streams are high, with the ultrapure water addition likely to cause only minimal effects to the microbial communities that tend to be associated with sediment particles and so likely only experienced some dilution of porewaters. This, combined with the large responses to temperature increases in the fine sediments that were observed in this study, indicate that the microbial communities were not greatly affected by osmotic shock or similar processes.” (Line 594).

(although I suspect the abundances in the water column would be much, much lower), a more robust experimental design would require the work be done more quickly, and using communities not subject to long and variable storage times.

I am also concerned re: variable oxygen in the systems based on comments in the response to line 357 and in comment 2.8. In response 2.8, re: water column oxygen – if sediments were 5cm deep, how deep was water? This will certainly have a large impact on oxygen – unless the water column is very deep.

The water column within the incubation jars was approximately 10 cm deep and was well mixed, reflecting the natural system in which the water column in the investigated streams is typically well mixed throughout the year, particularly in the thalweg. The depth of the water column impacting the oxygen content may be a concern for standing waters, however, in river systems stagnant waters only occur in oxbow lakes and deep pools, which were not considered here.

Additional concerns are -- the experiment includes ebullition – which in some ways is a benefit; however, once the CH₄ is in a bubble, the CH₄ will be subject to little or no oxidation, affecting CH₄:CO₂ ratios. That means the mix of processes and phases will make it difficult to understand what is going on, or how to extrapolate it.

We agree with the Reviewer that once CH₄ is within a bubble it will be subject to little or no oxidation, which will affect CH₄:CO₂ ratios. We did not separate diffusion and ebullitive fluxes, however, as ebullition occurs in natural systems, the inclusion here in a controlled experiment allows a clearer analysis of total fluxes of greenhouse gases and CH₄:CO₂ ratios. This is especially beneficial given that in-situ studies may underestimate ebullition, and results in a more robust experimental design.

More minor comments -- Geological origin as related to biogeochemical properties was noted – can you articulate the specific properties that are important here? Alternatively, just noting the geologic origin is adequate if you can't go further – as written line 47 (re: biogeochemical properties) seems self evident.

The reference to biogeochemical properties has been removed from the sentence to address the Reviewer's point that this statement was self-evident.

“Geological origin was investigated as different geologies are expected to have different GHG production rates.” (Line 47)

Sorption/desorption – The other reviewer noted issues re: mass recovery. Is the magnitude of potential impact expected to be on the order of 1% or less? If so, I think this is OK – but this methodological issue requires a more detailed discussion re: justifying why the mass recovery experiments aren't needed, and why what we know suggests the implication for the results are limited. I think this is likely the case, but this is a significant issue that merits further discussion.

The Reviewer refers to Reviewer 2's comments on resazurin (raz) and resorufin (rru) sorption differing between fine and coarse sediments, and sediments of different geological origin in the previous round of reviews. We addressed this comment directly to the Reviewer in the response to Reviewers without altering the manuscript. We have now included a discussion of our response while discussing the Rru/MMA results within the revised manuscript.

“Raz and Rru are known to sorb to sediments, which bears the risk that MMA could have been underestimated during these experiments, and all previous applications of the tracer. However, as such potential underestimation would have occurred throughout all investigated sediment types, it is expected that the impact on the interpretation of the results would be minimal. Additionally, it is expected that sorption would be slightly greater, and hence the underestimation greater, in the fine sediments. As the fine sediments had the greatest MMA, the potential underestimation in those sediments would not affect the conclusion that metabolism was greatest in the fine sediments. It is also possible that the sorption and mass recovery of Rru differed between the sediment types and geological origins, however, based on previous research and the relatively low organic matter contents across all sediment types in this study, this is expected to have a negligible effect.” (Line 113)

Re: response to line 281, I still disagree. Much of the GHG literature discusses productivity gradients with respect to CO₂ and CH₄ flux. The actual text is something where a small change might make it more acceptable. E.g., just note the fact it is important to consider streambed respiration in CO₂ production – rather than highlighting a research gap that I’m not sure exists.

The text has been altered to reflect the Reviewer’s suggestion:

“Our results indicate the importance of considering streambed respiration and subsequent CO₂ production, in C fluxes from streams and rivers, as this may alter the temperature-dependence of CO₂ emissions.” (Line 383).

Reviewer #2 (Remarks to the Author):

Dear editor and authors,

I have now reviewed the resubmission of the manuscript NCOMMS-17-16889A and I believe that it has improved to the point to recommend for publication. The authors have satisfactorily addressed all my comments. I especially value the new statistical approach and the streamline of the results section. The readability of the manuscript has substantially increased as a result.

Reviewer #3 (Remarks to the Author):

In the revised version of the manuscript about green house gas emission from different sediments of small streams the authors improved the statistical methods, the figures and the reading flow of the results and discussion section as suggested. I have no more concerns and recommend publishing this interesting and timely study in Nature Communications. There are two minor changes in the Material section highlighted in the attached pdf file.

These changes have been addressed by altering the text as suggested by the Reviewer.

Please see the changes on lines 410, 463 and 465.

We also made some minor changes to the statistical analysis as follows:

In addition, we have repeated the estimation of the statistical models in Tables S1-S3 using the residual maximum likelihood estimator rather than the maximum likelihood estimator. We have noted this on Line 464. This switch is adopting an even more cautious approach and is guarding against any risk that the maximum likelihood estimator leads to biased estimates of the variance for small sample sizes. The switch leaves the estimated values unaltered in Tables S1-S3, but the p-values have changed. The changes to the p-values do not substantially alter our findings.

We have altered the text in the Results section to more clearly convey how the statistical tests described in Section 2.7 relate to significant treatment effects.

References

1. Neal, C., Jarvie, H. P., Wade, A. J., Neal, M., Wyatt, R., Wickham, H., Hill, L. and Hewitt, N. The water quality of the LOCAR Pang and Lambourn catchments. *Hydrology and Earth System Sciences* **8(4)**, 614-635 (2004)
2. Krause, S., Tecklenburg, C., Munz, M. and Naden, E. Streambed nitrogen cycling beyond the hyporheic zone: Flow controls on horizontal patterns and depth distribution of nitrate and dissolved oxygen in the upwelling groundwater of a lowland river. *JGR:Biogeosciences* **118**, 54-67 (2013)